# Mesenchymal stem cell-derived extracellular matrix for musculoskeletal tissue regeneration
Shuqing Lv [1,5], Jia Wang[1,5], Jianan Chen[1], Yunyuan Yu[2], Xinying Huang[1], Gang Zhao [3 ✉], Xinfeng Zhou[1 ✉] & Yong Xu [1,4 ✉]

Mesenchymal stem cell-derived extracellular matrix (mECM) is increasingly recognized in tissue regeneration due to its high biocompatibility, controllability, and customizability. In musculoskeletal diseases, mECM provides a 3D scaffold mimicking the natural cellular environment and contains bioactive components regulating cell behavior and fate to promote tissue regeneration and repair. This review summarizes the preparation methods and composition of mECM, its effects on regulating cell behavior, and its applications in bone, cartilage, muscle, nerve, and blood vessel repair. It also analyzes the potential mechanisms of mECM's effects and identifies key challenges to be addressed prior to clinical translation, outlining future development directions.

Musculoskeletal disorders (MSK) demonstrate exceptionally high global prevalence, exerting substantial impacts on individual health and socio-economic frameworks. Current epidemiological projections indicate a substantial increase in MSK disorders among adolescents and young adults by 2050[1]. Furthermore, the challenges associated with musculoskeletal system repair and maintenance are intensifying alongside global population aging[2–5]. Within current therapeutic paradigms for MSK disorders, conventional strategies, such as pharmacological interventions and surgical procedures, primarily provide symptomatic management but remain inadequate for addressing critical-sized tissue defects. Clinical applications of autografts and allografts face inherent limitations, including donor site morbidity, immune rejection risks, and susceptibility to infections[6]. Consequently, accelerating the development of targeted therapeutic strategies to enhance musculoskeletal regeneration has emerged as a critical priority in translational medicine.

In the latest theoretical framework, "changes in the extracellular matrix" has been established as a core biomarker of aging[7]. The characteristic changes in ECM that occur with age (such as decreased viscoelasticity[8]) directly lead to the degeneration of the musculoskeletal system, thereby affecting the tissue repair ability[9]. Thus, a key challenge in developing ideal biomaterials is replicating the physicochemical properties and complex biophysical features of the natural ECM, which are essential for functional tissue repair in regenerative medicine. In this context, decellularized extracellular matrix (dECM) is emerging as a highly promising material for

facilitating effective regenerative processes in musculoskeletal tissue engineering. Biomaterials derived from dECM can support specific cell types and initiate intrinsic regenerative processes by mimicking the native tissue microenvironment.

Although dECM from various cellular and tissue sources has been extensively reviewed for tissue engineering applications[10–12], mECM has garnered particular interest for its potential in developing acellular transplantation biomaterials. The inevitable senescence of seed cells during in vitro expansion and the harsh post-implantation environment have significantly hindered the advancement of tissue engineering technologies. However, growing evidence suggests that mECM does more than offer a novel platform for in vitro cell expansion it also enhances the survival of transplanted cells in vivo[13]. Moreover, mECM materials exhibit anti-inflammatory[14–16], immunomodulatory, and pro-angiogenic properties similar to mesenchymal stem cells (MSCs), offering a promising approach for developing cell-free transplant biomaterials. Compared to tissue dECM or mature cell dECM, mECM offers several distinct advantages. For instance, mECM can be used autologously, reducing the risk of immune complications and donor site morbidity[17]. Importantly, mECM may enhance the differentiation potential of reseeded cells more effectively than ECM produced by mature cells[18].

Despite the widespread use of mECM in tissue engineering grafts, synthetic scaffold coatings, and pre-engraftment cell culture substrates, comprehensive reviews focusing on its role in musculoskeletal tissue

[1]Department of Orthopaedics, Orthopedic Institute, MOE Key Laboratory of Geriatric Diseases and Immunology, The First Affiliated Hospital of Soochow University, Suzhou Medical College, Soochow University, Suzhou, Jiangsu, China. [2]Department of Orthopaedics, The Third Affiliated Hospital of Soochow University, Changzhou, Jiangsu, China. [3]Department of Orthopaedics, Wuxi Ninth People's Hospital of Soochow University, Wuxi, China. [4]National Center for Translational Medicine (Shanghai) SHU Branch, Shanghai University, Shanghai, China. [5]These authors contributed equally: Shuqing Lv, Jia Wang. ✉e-mail: zhaogangmd@suda.edu.cn; xfzhou1112@suda.edu.cn; yxu1615@suda.edu.cn

engineering are scarce. This review begins by examining the decellularization methods, structural characteristics, and compositional properties of mECM, providing a foundational understanding of its biological and functional attributes. Following this, we delve into the effects of mECM on cellular behavior, particularly its influence on cell proliferation and senescence, which are critical determinants of regenerative potential. We then highlight current applications of mECM in musculoskeletal tissue regeneration, with a focus on both in vitro models and in vivo studies that demonstrate its translational potential. Finally, we explore the potential mechanisms through which mECM regulates cell behavior, discuss key challenges in the development of mECM-based biomaterials, and outline future directions aimed at advancing reproducible, scalable, and tissue-specific functional repair and regeneration strategies.

## Preparation of the mECM

### The structure and components of mECM

mECM is a three-dimensional (3D) matrix that supports cell attachment, migration, survival, and function. Under scanning electron microscopy (SEM), mECM appears as randomly arranged nanofiber bundles[19]. This porous architecture is essential for facilitating nutrient diffusion, oxygen transport, and waste exchange, while also providing sufficient space for cell infiltration during tissue regeneration. Atomic force microscopy studies further reveal that mECM exhibits tissue-specific mechanical properties, with Young's modulus varying from 0.1 to 10 kPa[20]; for instance, bone marrow-derived mECM is relatively stiffer due to dense collagen packing, whereas adipose-derived mECM is softer to match the mechanical microenvironment of soft tissues. Such structural features enable mECM to recapitulate the native tissue microenvironment and regulate cell behavior through mechanotransduction.

Studies have shown that mECM is primarily composed of collagen, fibronectin (FN), laminin, elastin, and other adhesive proteins, as well as various proteoglycans and hyaluronic acid[21]. Among collagens, type I collagen forms the fibrillar backbone to confer tensile strength[22], while type IV collagen assembles into sheet-like networks that constitute basement membrane-like domains, supporting cell polarization and adhesion[23]. FN contains conserved RGD (Arg-Gly-Asp) sequences that specifically bind to cell surface integrins (e.g., $\alpha5\beta1$), activating focal adhesion kinase-mediated signaling pathways to promote cell migration and survival[24]. Proteoglycans, such as decorin, regulate collagen fibril assembly and matrix integrity by enhancing molecular adhesion between aggrecan and collagen[25], while aggrecan forms large complexes with hyaluronic acid to endow mECM with compressive resilience. Hyaluronic acid, a key glycosaminoglycan, maintains matrix hydration and mediates cell proliferation via interactions with the CD44 receptor[26], which partly explains the elevated pro-regenerative capacity of young mECM with higher hyaluronic acid content[27].

When compared to tissue-derived decellularized ECM (tECM), mECM presents distinct structural and compositional characteristics, with each offering unique potential advantages (Table S1). tECM retains the complex, hierarchical architecture and tissue-specific bio-molecular composition of the native organ, providing a holistic microenvironment that can be difficult to fully replicate in vitro[28]. It is important to note that advancements in decellularization technologies, such as the use of supercritical $CO_2$ and mild bio-detergents, have significantly improved the preservation of crucial ECM components while reducing immunogenic residues in tECM scaffolds[29]. The inherent complexity of tECM, which mirrors the in vivo niche, may indeed be critical for orchestrating complex regenerative processes.

In contrast, mECM lacks the macroscopic, organ-specific tissue hierarchy of tECM. Its structure is more homogeneous and originates from a single, defined cell population. This defined origin is the source of its potential advantages, including a minimized risk of immunogenicity (particularly for autologous applications) and a high degree of controllability and customizability. The composition, stiffness, and bioactivity of mECM can be tailored by pre-conditioning the MSCs during the deposition phase. However, whether this tailored homogeneity is superior to the inherent

complexity of tECM for clinical outcomes lacks long-term translational data. Ultimately, the choice between mECM and tECM may be application-dependent, revolving around the specific trade-off between a highly complex, native microenvironment (tECM) and a well-defined, tunable, and potentially patient-specific platform (mECM).

## Decellularizing the mECM

Decellularization is the process of removing cells and cellular debris from tissues or cell cultures, leaving behind the extracellular components[30]. Due to immune responses, grafts from allogeneic or xenogeneic sources often have limited efficacy. For example, implanting allogeneic chondrocytes in cartilage defects can trigger varying degrees of immune rejection[31,32]. Residual cellular components, particularly DNA, are considered major antigens that can elicit immune responses in both allogeneic and xenogeneic transplants[33]. Additionally, for xenogeneic tissues (e.g., from porcine or bovine sources), the α-galactoside (α-Gal) epitope on cell surfaces is another predominant xenoantigen that triggers a potent immune reaction in human recipients. DNA and the α-galactoside (α-Gal) epitope on cell surfaces are considered major antigens that can elicit immune responses. Recent studies, such as those by Li et al., have shown that removing α-Gal antigens from decellularized porcine nerve matrices can significantly reduce immune responses in human hosts[34].

Unlike cells, ECM components are highly conserved across species and generally exhibit good immune tolerance[35]. Therefore, effective decellularization is crucial for reducing graft antigenicity, minimizing host inflammatory and immune responses, and enhancing material efficacy. Various decellularization methods have been reported, broadly categorized into physical and chemical/biological approaches[30,36–38]. Physical methods include freeze-thaw cycles, agitation, pressure gradients, and supercritical fluids, while chemical/biological methods involve acids, bases, detergents, alcohol, enzymes, and chelating agents. Physical stresses, such as freeze-thawing and osmotic pressure differences, can lyse cells without significantly disrupting tissue ultrastructure. However, physical methods alone often fail to completely remove residual debris and genetic material, necessitating combination with chemical or enzymatic treatments[39,40] (Fig. 1 and Table S2).

For mECM acquisition, chemical and biological decellularization methods are commonly used. Studies have identified two main approaches: nonionic detergents or chelating agents[30]. The nonionic surfactant Triton X-100 demonstrates selective lysis efficiency in decellularization protocols. It effectively removes cellular components while retaining secreted ECM proteins and matrix-associated components[41]. Combining nucleases, such as DNase and RNase, can further eliminate residual genetic material like DNA[15,35]. Chelating agents, represented by ethylenediaminetetraacetic acid (EDTA) and ethylene glycol tetraacetic acid (EGTA), function by sequestering divalent cations (e.g., $Ca^{2+}$, $Mg^{2+}$) critical for maintaining cell membrane integrity and intercellular junctions[41]. This cation depletion disrupts the structural stability of cell membranes, leading to gradual cell lysis, while their mild chemical nature minimizes damage to sensitive ECM components (e.g., collagen triple helices, glycosaminoglycan chains) that are vital for mECM's biomimetic activity[42]. For instance, EDTA is frequently used to loosen cell adhesion to the ECM scaffold by chelating $Ca^{2+}$ in focal adhesion complexes, and it is often combined with nucleases (DNase/RNase) to further eliminate residual genomic DNA and RNA—similar to the synergistic use of nonionic detergents with nucleases. Notably, EGTA exhibits higher selectivity for $Ca^{2+}$ over $Mg^{2+}$, making it preferable when preserving $Mg^{2+}$-dependent ECM enzymes (e.g., lysyl oxidase) is required for subsequent tissue regeneration[41].

However, there is currently no standardized method for obtaining mECM. Optimal decellularization protocols must achieve complete eradication of immunogenic remnants while retaining critical structural and bioactive constituents of the native ECM. This biomimetic strategy maintains the 3D ultrastructural integrity and native micromechanical niche essential for regenerative processes[43]. Therefore, the specific decellularization method may need to optimized according to the stem cell type, cell density, and desired ECM thickness[44].

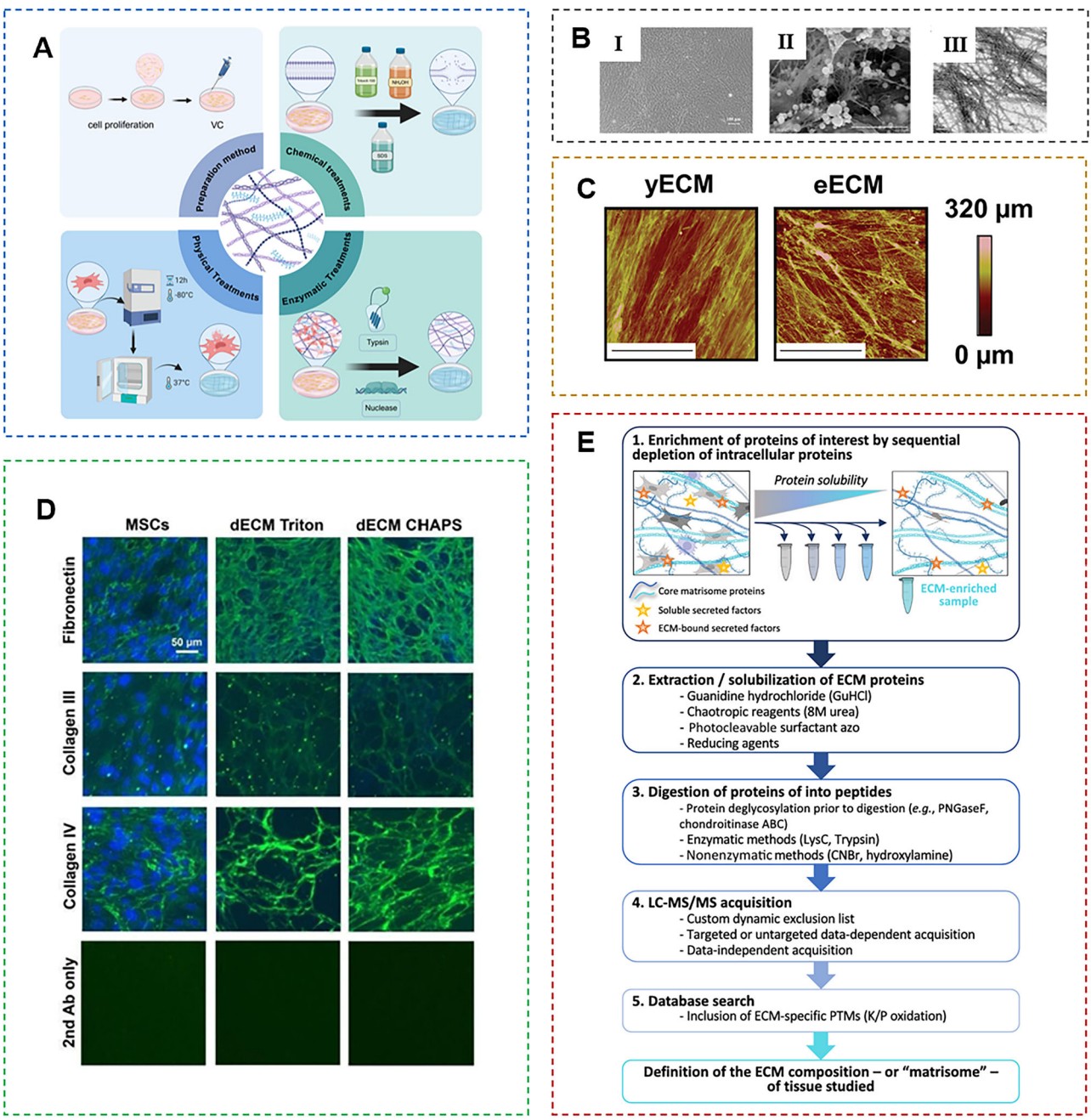

**Fig. 1 | Preparation and characterization of mECM. A** The method for preparing extracellular matrix includes physical method, chemical method, and biological method. Figure created with Biorender.com. **B** Characterization of mECM I phase-contrast microscopy images and II SEM and III TEM images of the BM-MSCs ECM[138]. **C** Atomic force microscopy (AFM) of young and elderly mECM.[139] **D** Immunofluorescence comparison before and after decellularization[140]. **E** Overview of ECM proteomics workflows[134].

## Characteristic differences of mECM from different sources of MSCs

ECM produced by MSCs from different sources exhibits significant differences in protein composition, growth factors, and immune regulation, giving each source unique advantages in tissue engineering and regenerative medicine applications. Ragelle et al. discovered that the mECM matrices deposited by bone marrow-derived mesenchymal stem cells (BM-MSCs) and adipose-derived stem cells (AD-MSCs) exhibit distinct matrisome profiles. Although ECM from different stem cell sources shares a subset of common proteins, their composition is markedly distinct from that of ECM derived from somatic cells. For example, BM-MSCs ECM is rich in factors related to the bone marrow microenvironment (such as CXCL12 and S100 proteins); AD-MSCs ECM specifically expresses tendinin XB and

connective tissue growth factor. The overall collagen content of neonatal fibroblasts ECM is relatively low, but it is rich in fibrinogen 2 and proteins related to the TGF-β and WNT signaling pathways (such as LTBP4, WNT5A). These specific protein composition characteristics reflect the functional specificity of their original tissues and differentially regulate the transcriptome and behavior of cells, emphasizing the importance of considering ECM sources in tissue engineering for precisely simulating the in vivo microenvironment[21]. For example, BM-MSCs-derived ECM has shown exceptional performance in bone tissue engineering. Feng et al. demonstrated that rat BM-MSCs-derived ECM tablets promote bone integration[45]. AD-MSCs-derived ECM has proven effective in soft tissue repair and cardiovascular disease treatment. Gangadaran et al. found that AD-MSCs-derived extracellular vesicles contain pro-angiogenic proteins,

such as IL-8, CCL2, and vascular endothelial growth factor (VEGF), which promote angiogenesis in vitro and in vivo[46]. Additionally, AD-MSC-derived ECM hydrogels have been shown to enhance cartilage tissue regeneration[47]. Umbilical cord mesenchymal stem cell (UC-MSCs)-derived ECM has demonstrated excellent performance in nerve regeneration. Xiao et al. identified UC-MSCs-derived ECM components using immuno-fluorescence and confirmed that it is rich in laminin, FN, and types I, IV, and X collagen, which play crucial roles in peripheral nerve regeneration. The 3D structure of UC-MSCs ECM, with its numerous micropores, provides a foundation for modifying neural conduits[48].

ECM generated from MSCs of different origins holds broad application potential in tissue engineering and regenerative medicine. Further research and optimization of mECM preparation methods will enhance its clinical efficacy.

## The effect of mECM on cell behavior
### The influence of mECM on cell proliferation
mECM provides an excellent platform for the in vitro expansion of cells. Numerous studies have demonstrated that mECM significantly enhances the proliferative capacity of cultured cells while maintaining their inherent characteristics[49–51]. For instance, autologous cell-derived mECM substantially improves the proliferation of BM-MSCs, UC-MSCs and SM-MSCs[50,52,53]. Moreover, mECM enhances differentiation of adipose stem cells from the infrapatellar fat pad toward chondrogenesis[54]. Notably, mECM exhibits dual functionality by augmenting the proliferative potential of SM-MSCs while simultaneously establishing a biomimetic micro-environment conducive to the survival and functionality of terminally differentiated cells[15,55]. For example, Yan et al. found that chondrocytes cultured on mECM exhibited a significantly higher proliferation rate compared to those grown on tissue culture polystyrene (TCPs). Further studies indicated that chondrocytes expanded on mECM possessed stronger anti-inflammatory capabilities than those expanded on TCPs[15]. Therefore, mECM holds promise as an ideal platform for the in vitro expansion of seed cells. It is worth noting that mECM's ability to promote cell proliferation is age-dependent. Comparative analysis by Li et al. demonstrated that while both fetal mECM and adult mECM stimulated robust proliferation of stem cells in vitro, fetal mECM exhibited markedly enhanced mitogenic potential[56]. Additionally, studies have shown that mECM can adsorb growth factors or hormones, further enhancing cell proliferation. This opens up broader prospects for the application of mECM in cell expansion[57].

### The influence of mECM on cell senescence
Cell senescence is a state of permanent proliferation arrest induced by various stress factors[58], accompanied by multiple phenotypic changes. The decline in proliferation capacity and weakened damage repair ability caused by cellular senescence have become significant limitations in the repair of bones and cartilage using seed cells, such as stem cells or chondrocytes. The cellular microenvironment is vital for regulating cell behavior, particularly senescence and the ECM is a crucial part of this microenvironment, influencing cell survival, migration, proliferation, and differentiation[59,60]. Studies have found that excessive $H_2O_2$ stimulation can trigger premature senescence in stem cells, while mECM can provide a protective effect by shielding stem cells from oxidative stress-induced premature senescence. Further research has revealed that the anti-aging effect of mECM is primarily related to the type I collagen component in the matrix, rather than FN[53].

Young mECM has been shown to have greater benefits in delaying cellular senescence. In one study, researchers cultured aged BM-MSCs on both aged mECM and young mECM. The results indicated that stem cells cultured in the young mECM environment exhibited enhanced self-renewal and osteogenic differentiation capabilities compared to those cultured in the aged ECM environment. This improvement was accompanied by increased telomerase activity and higher levels of ATP production. In addition, ECM derived from young MSCs also showed superior role in promoting cell proliferation and multiline differentiation of BM-MSCs[61].

In addition to regulating stem cell fate, mECM can partially reverse the aging process of mature cells, such as chondrocytes. Researchers like Pei et al. have found that ECM derived from synovium-derived MSCs significantly delays replicative senescence and dedifferentiation of chondrocytes while markedly enhancing their redifferentiation ability. Furthermore, the expression level of CD90, a "stemness" marker, significantly increases during amplification on mECM[62]. In summary, mECM has a significant impact on cellular senescence and can partially reverse the aging process.

### The mechanism of mECM affecting cell behaviors
The mechanism of mECM regulating cell behavior has not been clarified. mECM is a complex mesh scaffold composed of glycosaminoglycan, proteoglycan, collagen and non-collagen, and its matrix components affect cell adhesion, migration, proliferation and differentiation[63]. Zhou et al. showed that collagen type I (COL I) is pivotal in regulating cell senescence. When using $H_2O_2$ to induce premature aging, the proportion of senescent human umbilical cord mesenchymal stem cells on COL I was significantly lower than those cultured on TCPs or FN, but its anti-aging effect was weaker than that of intact mECM matrix[53]. Interestingly, osteoclast differentiation was almost unaffected by COL I and was mainly related to FN[64]. Therefore, the regulation of mECM on different cell behaviors may be mainly mediated by different matrix components. In addition, the mechanical properties of the ECM also play an important role in regulating cells. SEM data results showed that SM-MSCs growing on the mECM substrate with 3D structure showed good fibroblast-like cell shape, rather than the wide and flat cell shape shown on a smooth flat plastic flask[65]. The results of Li et al. showed that compared with adult mECM, fetal mECM showed lower hardness, and the hardness of amplified SM-MSCs was consistent with the hardness of the medium. Therefore, the authors hypothesized that the young mECM had better ability to promote cell proliferation and chondrogenic differentiation than the old mECM, relative to old mECM might be related to its lower elasticity[56].

In addition to the influence of mECM matrix composition and mechanical properties, mECM can also activate several important intracellular signaling pathways (Fig. 2 and Table 1). Signaling pathways are crucial for regulating cellular behavior. Silencing information regulator type 1, a class III histone deacetylase, plays a crucial regulatory role in multiple physiological processes, including cellular aging dynamics, metabolic homeostasis maintenance, and programmed cell death mechanisms. mECM has been reported to activate SIRT1-dependent signaling pathways, increase intracellular anti-senescence and antioxidant enzyme levels[53,66], and enhance anti-inflammatory and anti-aging capabilities of cells[15]. In addition, the blocking of the NF-κB signaling pathway has been shown to be associated with mECM's inhibition of osteoclast formation[64]. Deng et al. showed that UC-MSCs-ECM activated the integrin signaling pathway in a TGF-β1-dependent manner, which promoted the migration of M2 macrophages and induced the repolarization of M1 macrophages into M2 macrophages[14]. The regulatory role of mECM on cell behavior still needs to be further elucidated.

## Application of mECM in musculoskeletal tissue regeneration
In the quest for biomaterials suitable for treating musculoskeletal diseases, mECM has drawn considerable interest due to its remarkable efficacy in regulating cellular behavior. Researchers have extensively tested various forms of mECM applications (Fig. 3). The research shows that mECM has great potential in different application forms. For example, studies have used allogeneic acellular BM-MSCs sheets to prepare biological extracellular matrix scaffolds[16]. Such mECM sheets, obtained through decellularization, provide a surface that mimics the in vivo environment, promoting cell attachment, proliferation, and differentiation, and achieving osteochondral reconstruction. Chiang et al. developed mECM scaffolds derived from 3D acellular mesenchymal stem cell spheres. This decellularized 3D mECM structure, produced under macromolecular crowding, fully preserves the microstructural ECM components and has a high growth factor retention

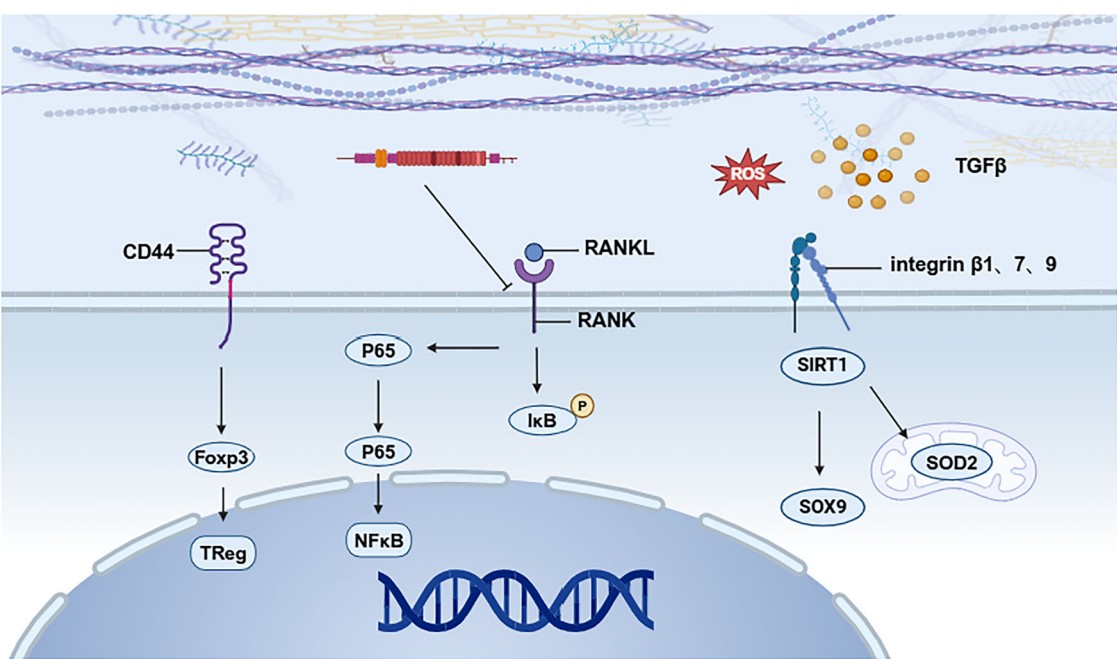

**Fig. 2 | The possible mechanisms of mECM affecting cell behaviors.** mECM can regulate cellular processes, including proliferation, senescence, inflammation and differentiation, by activating key pathways.

## Table 1 | The possible mechanisms of mECM affecting cell behaviors

| Function | Involved ECM components | Involved signaling pathway |
|---|---|---|
| Antioxidation | / | SIRT1[66] |
| Anti-inflammation | / | SIRT1[15] |
| Immunomodulation | Hyaluronic acid | CD44[135] |
| | TGF-β1 | Integrinβ7, integrinβ9, and integrinβ1[14] |
| Anti-senescence | Collagen type I | SIRT1[53] |
| Anti-osteoclastogenic property | Fibronectin | NF-κB[64] |
| Proliferation and maintenance of stemness | Non-collagenous proteins[136] | / |
| Angiogenic differentiation | Angiogenic-related cytokines[137] | / |

rate, promoting angiogenesis[67]. Additionally, mECM can be prepared into mECM powder using freeze-drying technology and combined with microspheres prepared by microfluidic technology through chemical crosslinking. This approach effectively delivers various inducible biological activity factors in mECM while promoting cell survival and growth due to the large adhesion surface and good porosity of the microspheres. For example, one research team achieved bone defect repair in aging rats by combining young mECM freeze-dried powder with hydrogel microspheres[61]. Another example is Antich et al., who digested freeze-dried mECM powder in pepsin solution to prepare ECM hydrogels with different concentrations. These hydrogels can induce chondrogenesis of MSCs without the need for complementary factors and form tissues similar to hyaline cartilage after in vivo implantation[47]. These results demonstrate that mECM and its derivatives have great potential in tissue engineering, providing effective solutions for the repair of various tissue defects. By continuously optimizing and innovating the application form of mECM, it can better promote the development of biomedical research. In the following

sections, we will focus on mECM applications in bone, cartilage, muscle, tendon, ligament, nerve, and blood vessel regeneration.

## The application of mECM in bone regeneration

The constant interplay between bone formation and resorption, combined with the development of new blood vessel networks, is a critical factor in the successful regeneration of bone tissue. The effects of mECM on bone formation have been extensively studied in vitro. Many studies have shown that MSCs cultured on autologous mECM exhibit enhanced osteogenic differentiation potential and reduced intracellular reactive oxygen species (ROS)[50,66,68]. Interestingly, mECM can support osteogenic differentiation of stem cells even without the addition of dexamethasone[69], a common osteogenic induction additive. MSCs that were expanded on mECM and transplanted subcutaneously into nude mice retained their enhanced osteogenic ability in vivo[70]. Additionally, the effects of mECM on bone resorption and neovascularization have been preliminarily studied. Our group first reported that mECM significantly inhibits osteoclast formation by reducing intracellular ROS[64]. Concurrently, mECM stores a large number of angiogenic-related factors and supports stem cells in secreting more VEGF, thereby enhancing the angiogenic potential of MSCs[71,72]. These studies highlight the importance of mECM in regulating bone regeneration.

The impact of mECM on bone formation has been evaluated both in vivo at ectopic and orthotopic sites. In a subcutaneous ectopic bone formation model, mECM-modified polymeric materials effectively spurred mineralized matrix formation, as proven by positive bone sialoprotein immunostaining results[73]. Feng et al. demonstrated that BM-MSCs cultured on mECM sheets showed significantly improved adhesion and proliferation, and mECM sheets promoted the expression of osteogenesis-related genes. In vivo, mECM sheet-coated implants exhibited superior new bone formation compared to uncoated implants[45]. Larochette et al. found that mECM can also deliver bone morphogenetic protein-2 (BMP-2)[74], an important bone-inducing growth factor, in addition to serving as a scaffold coating. Furthermore, mECM sheets produced from osteogenic cell sheets —a construct where MSCs are cultured to form a sheet rich in osteoblasts, native ECM, and growth factors-maintained their structural integrity and contained abundant bone-induced growth factors after freeze-thaw cycling in liquid nitrogen. Notably, direct use of these mECM sheets[75], even without scaffolds, significantly promoted bone regeneration in rat nonunion models.

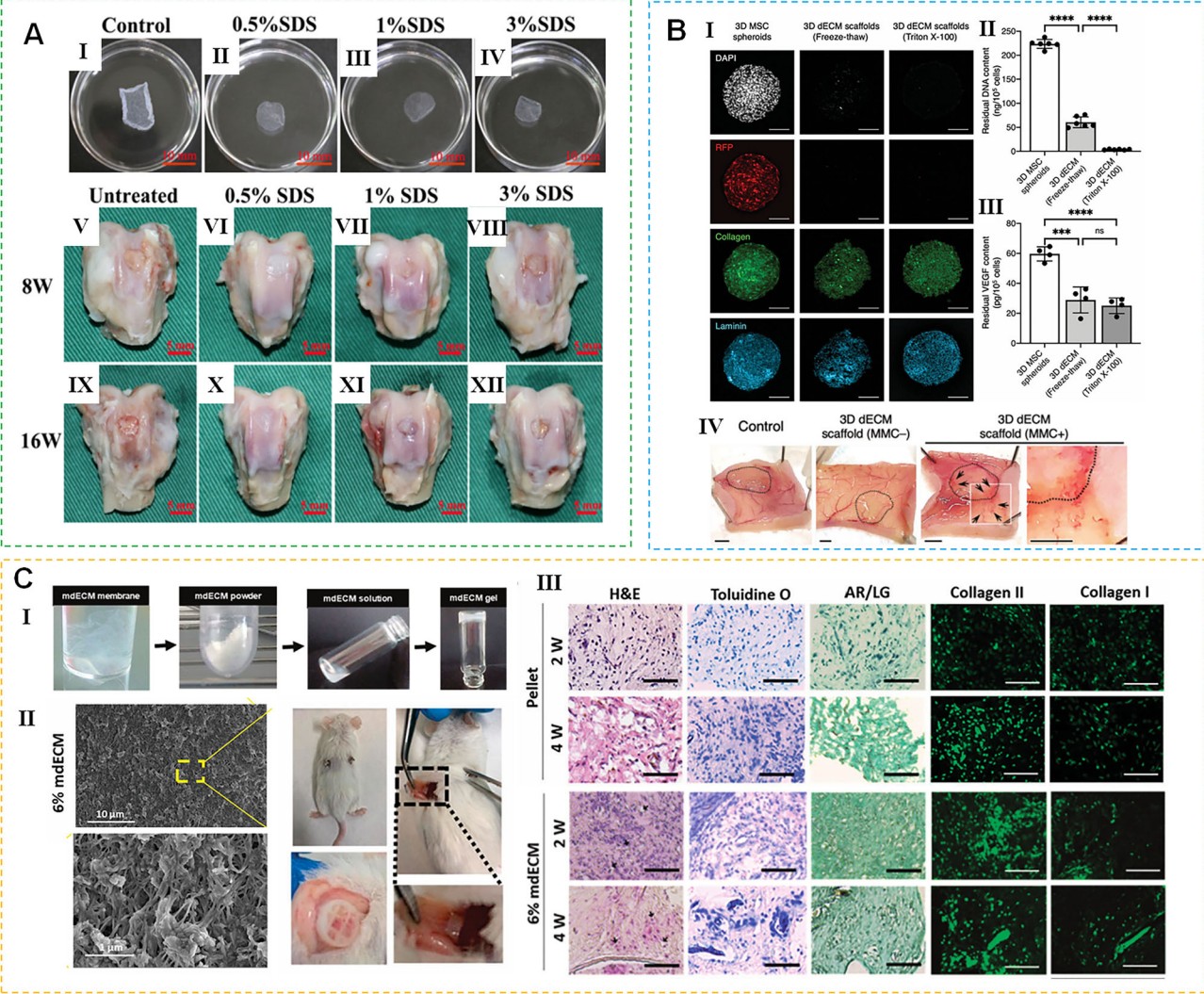

**Fig. 3 | Schematic of different application forms of mECM. A** Application of mECM as sheets I–IV characteristics of mECM sheets. V–XII mECM sheets promoted the regeneration of osteochondral defects in rabbits[16]. **B** Application of mECM as 3D spheroids. I Representative fluorescence images of 3D mECM spheroids before and after decellularization. II The residual DNA content within mECM is almost entirely depleted. III Significant amounts of soluble factors remain in the mECM spheroids. IV mECM spheroids promote angiogenesis[67].
**C** Application of mECM as biomimetic hydrogel. I Scheme demonstrating the fabrication of mECM hydrogel. II SEM of mECM hydrogel. III mECM hydrogels for in vivo induction of cartilage-like tissue[47].

Therefore, mECM represents a promising material for bone tissue engineering, whether as a scaffold coating or a standalone scaffold (Fig. 4 and Table S3).

## The application of mECM in cartilage regeneration
Cartilage, as an avascular tissue, presents significant challenges for regeneration due to limited access to nutrients and circulating progenitor cells[76]. mECM provides a suitable 3D microenvironment for chondrogenesis. Numerous studies have reported that mECM derived from both autologous and allogeneic stem cells significantly improves the chondrogenic differentiation potential of stem cells[49,50,54,77]. Moreover, Cristina Antich, et al. found that even without exogenous growth factors, mECM derived from autologous AD-MSCs could induce chondrogenic differentiation[47].

To assess chondrogenic potential, SM-MSCs that had been expanded on mECM were first differentiated into micropellets in vitro. These cells were then injected into cartilage defects in the knee joint. The beneficial effects of mECM pretreatment were evident in both settings: the micropellets showed enhanced chondrogenesis in vitro, and the injected cells generated superior repair tissue in vivo. After three months, the new tissue generated by injecting mECM-pretreated SM-MSCs expressed more glycosaminoglycan (GAG) and type II collagen than tissue generated by TCP-

amplified cells[52]. Additionally, mECM is emerging as increasingly vital in maintaining chondrocyte phenotypes, which is critical for autologous chondrocyte transplantation (ACI)[62]. When articular chondrocytes were cultured on SM-MSCs mECM, they showed delayed dedifferentiation and enhanced redifferentiation potential. Compared to chondrocytes cultured on TCPs, those cultured on BM-MSCs mECM maintained a better chondrogenic phenotype and reduced expression of hypertrophy markers for example type X collagen and alkaline phosphatase (ALP)[78]. However, Thakkar et al. found that mECM deposited by BM-MSCs did not restore the chondrogenic phenotype of osteoarthritic chondrocytes[79]. It should be noted that the mECM in this study was derived from BM-MSCs of a 75-year-old female patient, and the aging mECM may have limited therapeutic effects. Therefore, the impact of mECM on osteoarthritis chondrocytes requires further investigation.

Research has also investigated how mECM from various stem cell sources influences chondrogenesis. AD-MSCs cultured on AD-MSCs ECM showed comparable chondrogenic potential to those on SM-MSCs ECM[54]. Li et al. compared the effects of ECM deposited by SM-MSCs of different ages on the chondrogenic differentiation potential of adult stem cells and found that fetal SM-MSCs ECM was superior to adult SM-MSCs ECM in promoting chondrogenic differentiation. The authors suggested that these

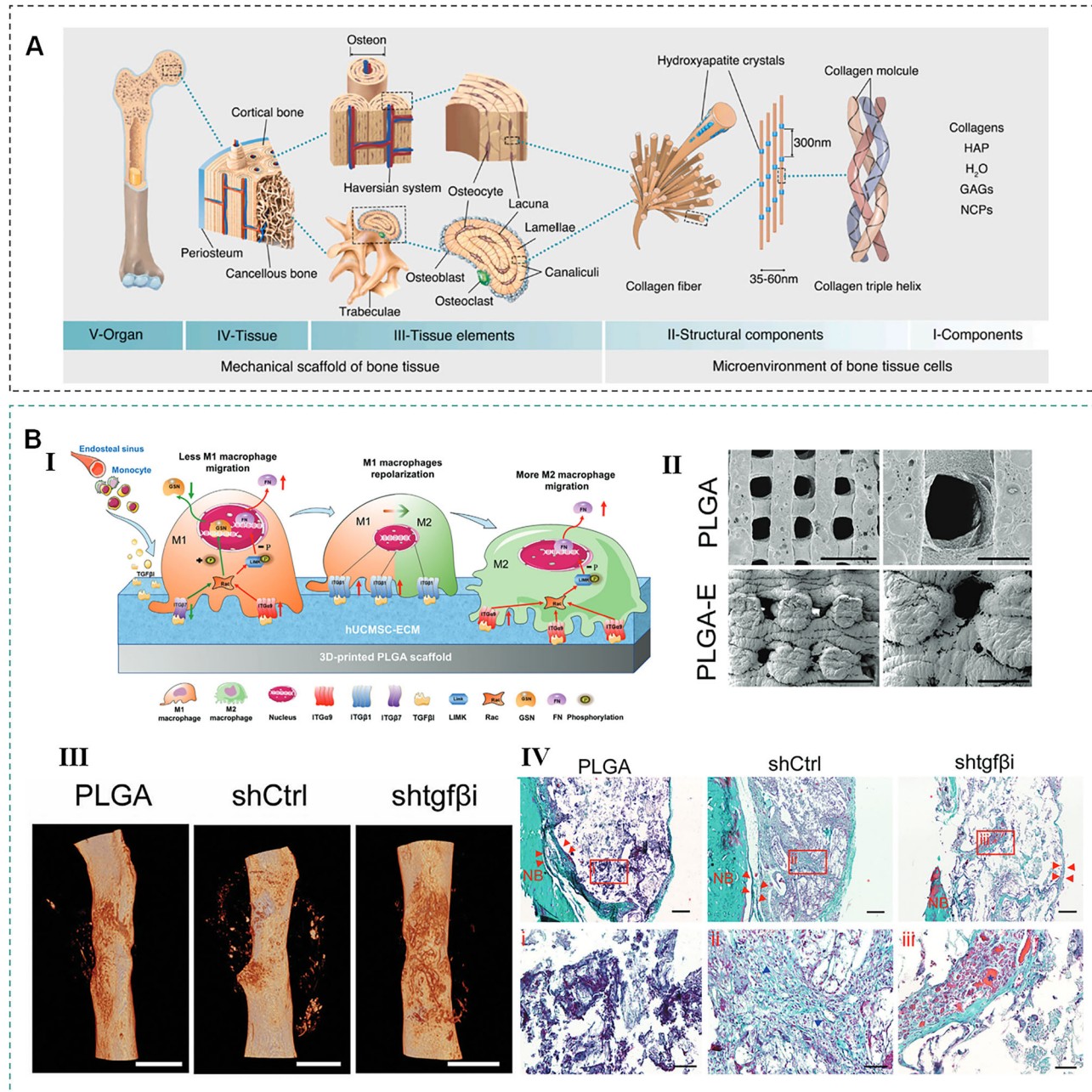

**Fig. 4 | Application of mECM in bone. A** Bone hierarchical structure[141]. **B** mECM modified PLGA scaffold promotes bone regeneration. I Potential mechanism of immunomodulation of PLGA-mECM. II The SEM images of scaffolds. III 3D view images of bone regeneration of PLGA-mECM. IV Histological assessment of regenerated tissue in the PLGA-mECM[14].

results may be related to the unique protein composition and low elasticity of young mECM[56]. Moreover, mECM deposited by MSCs during the early stages of chondrogenic differentiation (e.g., after ~1 week of induction), which mimics early chondrogenesis, promotes chondrogenesis, while mECM deposited during the late stages (e.g., after ~3 weeks of induction), which mimics late chondrogenesis, inhibits it[80]. Wang et al. used the SV40 large T antigen to transduce autologous infrapatellar fat pad stem cells (IPFSCs) to create immortalized stem cells and deposited ECM. Compared to control cartilage particles, replicative aging IPFSCs amplified on ECM deposited by immortalized stem cells produced larger cartilage particles with stronger sulfated GAG and type II collagen staining. This immortalization strategy, achieved through genetic modification of stem cells, may expand the sources of mECM and further advance cartilage tissue engineering[81].

mECM has also been successfully applied in cartilage repair in vivo. Tang et al. fabricated 3D porous scaffolds from ECM deposited by autologous bone marrow stem cells using cross-linking and freeze-drying techniques. In subcutaneous grafts in nude mice, mECM-made bioactive scaffolds induced the formation of thicker cartilage tissue compared to collagen scaffolds[82]. Tang et al. further demonstrated that implanting mECM scaffolds at cartilage defect sites enhanced the effect of bone marrow stimulation on cartilage repair[83,84]. In rabbit cartilage defects, bioactive scaffolds with mECM showed better cartilage regeneration than micro-fractures alone (Fig. 5 and Table S4)[85]. In addition to being used to make biological scaffolds, mECM-derived hydrogels also exhibit excellent bio-compatibility and the ability to induce chondrogenesis. In nude mice, mECM hydrogel showed good tissue integration and induced the formation of mature cartilage matrix in vivo[47].

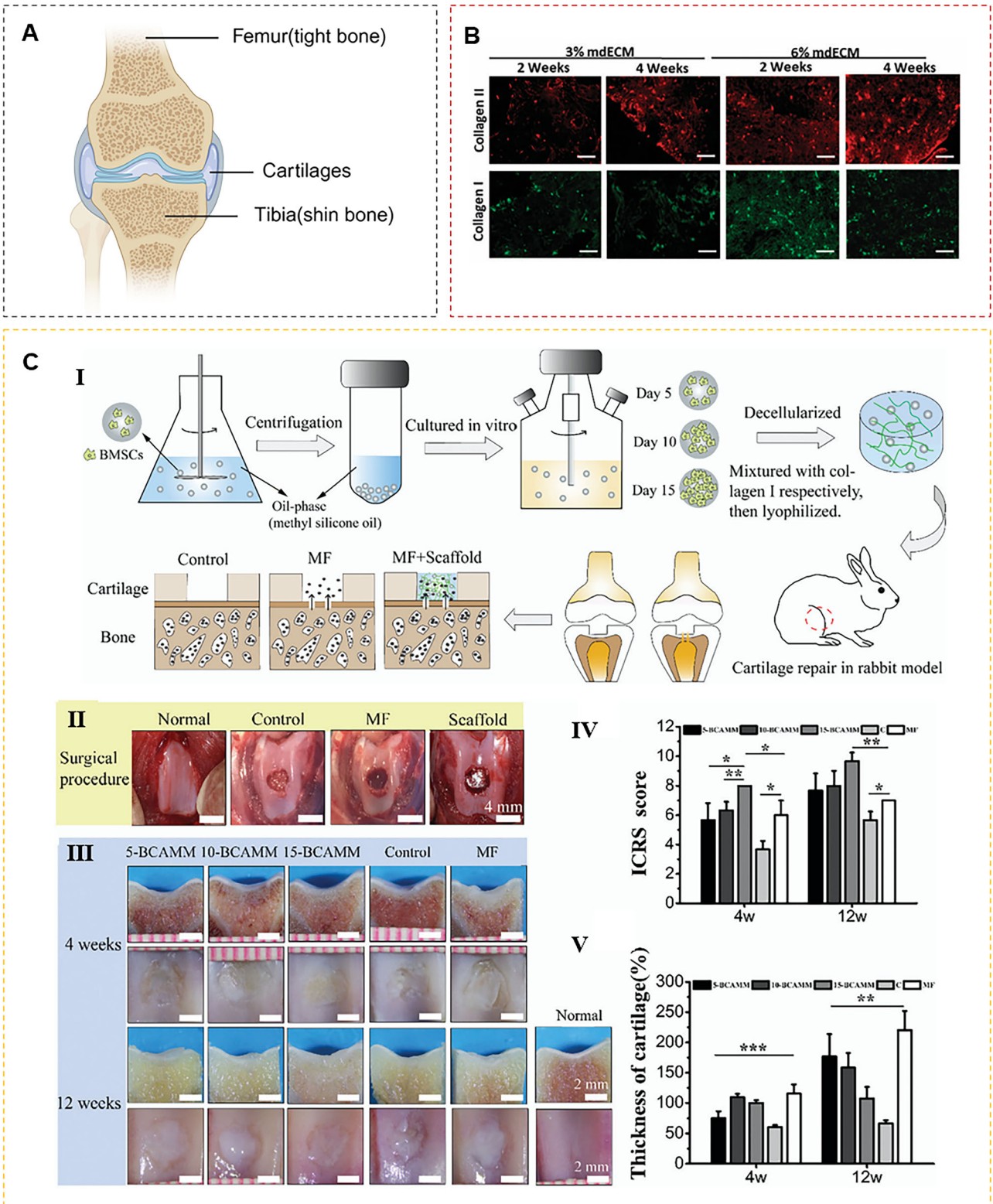

**Fig. 5 | The application of mECM in cartilage regeneration. A** Cartilage hierarchical structure. **B** mECM environment promotes collagen production of MSCs[47]. **C** mECM enhances the repair of articular cartilage defect. I Schematic illustrating scaffold fabrication and surgical implantation protocols. II mECM scaffolds were used to fill the chondral defects created on the patellar trochlear groove. III Gross and section morphology after 4 and 12 weeks. IV Macroscopic evaluation. V The thickness of the regenerated tissue[85].

## The application of mECM in muscle regeneration

Skeletal muscle consists of approximately 600 muscle bundles[86]. These bundles generate force through contraction and are the primary drivers of the musculoskeletal system. Resident stem cells, known as muscle satellite cells, are activated upon injury to rebuild damaged tissue. However, the regenerative capacity of skeletal muscle is insufficient to address critical-sized injuries or volumetric muscle loss (VML)[87]. In older individuals, surviving satellite cells often exhibit functional deficiencies[88]. Additionally,

senescent cells create a pro-inflammatory microenvironment and secrete senescence-associated secretory phenotype[89], a process that consequently diminishes the proliferative capacity of muscle satellite cells[90].

Current clinical treatments primarily focus on restoring irreplaceable motor function, with common surgical methods including scar tissue removal and autologous muscle transplantation[91]. However, these interventions often have limited efficacy and potential complications[92]. Biomaterials that promote cell adhesion and proliferation are ideal for aiding muscle tissue repair. Currently, acellular matrices from various sources have been applied to skeletal muscle regeneration with varying degrees of success[93]. For example, in 2010, VJ. Mase Jr. and his team developed a multilayer biological scaffold constructed from the extracellular matrix of pig intestinal submucosa. This scaffold was successfully implanted in the right thigh of a patient with severe muscle injury, leading to smooth recovery without complications. Four months later, the patient showed significant improvements in muscle function during isokinetic exercise tests, and computed tomography images confirmed the formation of new muscle tissue[94]. However, the regenerative ability of dECM scaffolds depends on attracting host cells, which often leads to disordered cell distribution in the VML region. This results in disorganized muscle fiber tissue and scar formation, making it difficult to align new tissue with the original healthy tissue, thereby limiting functional regeneration. In contrast, YJ Choi and colleagues achieved breakthroughs in treating VML injuries by constructing 3D cell scaffolds using extracellular matrix, achieving up to 85% functional recovery and significantly improving therapeutic outcomes[95]. Recently, Cassandra Reed et al. explored the use of muscle fibroblast-derived acellular matrix in rat tibialis anterior muscle injury model and found that its recovery results were superior to current tissue-derived ECM scaffolds[96].

Although various decellularized matrices have been utilized for skeletal muscle regeneration, most current studies still adopt a "cell-matrix co-delivery" strategy, where cells and matrices are implanted together into the injury site, without achieving standardized, independent separation and application of ECM[97]. Although this approach promotes functional recovery to some extent, it remains difficult to clearly distinguish whether the effects are attributable to the cells or the matrix, thereby limiting the systematic evaluation and standardized preparation of ECM as an independent biomaterial. Wang et al. demonstrated that injecting decellularized adipose-derived stem cell ECM fragments as a bulking agent into the urethra of a stress urinary incontinence rat model not only achieved physical bulking but, more importantly, recruited host cells and promoted local smooth muscle regeneration, significantly restoring urethral function. This highlights the great potential of ADSC-ECM in functional muscle regeneration[98]. Researches have shown that increasing the stiffness of hydrogels can effectively promote the regenerative ability of muscle stem cells[99]. Therefore, whether muscle defects can be repaired by increasing the stiffness of mECM is an important topic for further exploration. Given mECM's ability to simulate the natural cell growth environment and maintain cell "stemness"[100], its potential in promoting muscle regeneration is particularly promising. It is expected to provide new strategies and methods for treating large-volume muscle defects.

## The application of mECM in tendon/ligament regeneration

Tendons and ligaments are critical fibrous connective tissues in the human body[101]. Tendons connect muscles to bones, transmitting the force of muscle contraction[86], while ligaments connect bones, stabilize joints, and enable movement. Injuries to these structures, whether from overuse, trauma, or disease, can lead to pain[102], reduced function, and even disability[103]. Tendon and ligament injuries significantly impacting the quality of life for millions worldwide[104].

Because of their poor vascularity and limited regenerative capacity[105], tendon and ligament injuries often require surgical intervention[106]. However, current surgical methods, including autologous[107], allogeneic[108] and xenograft[108], are limited by donor shortages, immune[106] rejection[109], and poor clinical outcomes. Cryopreserved grafts often lack cell activity[110], leading to unsatisfactory functional recovery and long-term stability.

Additionally, the enthesis—the attachment point of tendons and ligaments to bone-is a structurally and functionally complex region[111], posing significant challenges for tissue regeneration.

Cells are the primary source of tissue immunogenicity[112]. Due to its complex structure, excellent biocompatibility[113], and abundance of bioactive factors, a cellular matrix[112,114] is considered a logical alternative for grafts. For example, porcine small intestine submucosa, porcine dermis, and human dermal tissue have been widely used for rotator cuff tears,[114]. Jiang et al. showed that ECM prepared by young tendon stem cells (TSCs) significantly promotes the proliferation and differentiation of elderly TSCs, reduces age-related β-galactosidase activity, and enhances the expression of stem cell markers in elderly TSCs, preserving their stem cell characteristics[115]. This suggests that TSC-derived ECM has great potential for tendon repair (Fig. S1).

mECM not only promotes tendon and ligament repair by providing necessary biological signals and structural support but also simulates the natural extracellular environment, attracting and activating surrounding stem cells and promoting their differentiation into tendon and ligament cells. This accelerates the healing process of injured tissues. For example, Ouyang Hongwei et al. used mesenchymal stem cell sheet technology to promote cell interconnection through self-generated matrix, simplifying the assembly process of cells in dense grafts and addressing the low cell attachment efficiency of traditional tissue engineering scaffolds[110].

Additionally, the use of acellular matrix helps reduce inflammatory responses and prevent scar tissue formation[116], which is essential for improving repair quality and restoring function. However, a single material class often fails to meet the complex requirements for biological activity and mechanical properties in tendon/ligament regeneration. In the future, further physical and chemical modifications[108] of mECM should be explored, or mECM should be used as a primary component of composite biomaterials to synergistically combine beneficial properties and achieve better repair outcomes.

## The application of mECM in nerve/vascular regeneration

The musculoskeletal system is rich in vascular networks and densely populated with sensory and sympathetic nerve fibers[117]. Blood vessels and nerves[118] are crucial[119,120] for the repair of musculoskeletal tissues. For instance, interrupted blood supply during a fracture can trigger osteonecrosis, characterized by extensive local cell death. Studies have found that peripheral nerve damage can delay fracture healing. During repair, sensory nerves communicate with osteoblasts through signaling pathway, promoting bone formation[121]. Simultaneously, growth factors such as VEGF and BMP-2 secreted by vascular endothelial cells activate osteoblasts and promote the differentiation of BM-MSCs into osteoblasts[122].

The muscular system is also rich in blood vessels and densely populated with sensory and sympathetic nerve fibers, which control muscle movement, transmit sensory information to the central nervous system, and regulate autonomic functions. Injuries to the musculoskeletal system often involve nerve fiber rupture, disrupting nerve signal transmission, and affecting muscle contractile function. Although autologous tissue transplantation is viewed as the optimal method for peripheral nerve repair, its clinical application is limited by donor shortages, tissue matching issues, and potential permanent nerve damage at the donor site. mECM has overcome these limitations due to its unique biochemical characteristics and structural layout, providing a finely regulated microenvironment for nerve regeneration. Research by Wang et al. confirmed that mECM derived from human BM-MSCs significantly promotes nerve repair, revealing its broad application prospects in accelerating peripheral nerve regeneration[123].

Since nerve regeneration is a slow process, single ECM components are often quickly absorbed and cannot provide long-term structural support. By combining ECM with polymer materials, this issue can be effectively addressed, providing a stable scaffold for nerve regeneration and maintaining long-term structural stability. Guan et al. extracted mECM from human UC-MSCs and modified it onto electrospun fibers, creating a microenvironment that provides both biochemical signals and physical

support, significantly accelerating nerve regeneration. Nanofibers prepared by electrospinning technology can simulate the oriented nano-layered structure of normal nerve matrix, which is helpful to achieve good directional arrangement of nanofibers[124]. It is worth noting that the hardness of this kind of mECM is similar to that of natural nerve tissue. When this mECM is attached to the surface of electrospun fibers, it can provide a suitable mechanical microenvironment for nerve regeneration (Fig. 6 and Table S5).

Blood vessels are indispensable for maintaining musculoskeletal homeostasis through coordinated regulation of cellular trafficking dynamics, oxygen-nutrient supply-demand coupling, metabolic waste clearance, and angiocrine-mediated tissue crosstalk[125,126]. Recent advances in extracellular matrix (ECM) engineering have revealed its critical influence on vascularization processes. Wang's team demonstrated innovative ECM modification strategies using urinary stem cell-derived ECM (USCs-ECM) to enhance porcine small intestinal submucosa (SIS) scaffolds. Their SIS + USCs-ECM composite exhibited remarkable angiogenic potential in vitro and accelerated cutaneous wound healing in vivo[127]. Extending this approach, the researchers developed extracellular matrix-synthetic polymer scaffold (ECM-SPS) by integrating endometrial epithelial and smooth muscle cell-derived ECM with polyurethane/SIS bilayers, achieving superior cellularization, vascularization, and immunomodulatory capabilities[128].

The vascular regulatory properties of ECM appear source-dependent. While chondrocyte-derived ECM (C-ECM) demonstrates anti-angiogenic effects[129], mECM exhibits potent pro-angiogenic characteristics (Fig. S2). Ma et al. revealed that hUMSCs-derived mECM enhances HUVEC angiogenesis via time-dependent activation of the integrin αVβ3/c-Myc/P300/VEGF axis[130]. This dual mechanism involves both direct delivery of angiogenic factors and paracrine stimulation of VEGF secretion from neighboring stem cells.

Notably, Carvalho et al. developed a synergistic approach through BM-MSCs/HUVEC co-culture (1:3 ratio), creating dECM scaffolds enriched with VEGF and FGF-2 that surpassed single-source mECM in angiogenic performance[131]. Emerging 3D fabrication techniques further amplify these effects-spheroid-derived 3D mECM scaffolds better recapitulate native ECM architecture, significantly enhancing endothelial cell dynamics and creating pro-angiogenic microenvironments through combined biochemical and topographical cues[67]. These advancements highlight the evolving paradigm of ECM engineering, where strategic source selection, cellular crosstalk modulation, and 3D structural bio-fabrication converge to optimize vascular regenerative outcomes.

## The clinical translation of mECM

For decades, ECM scaffolds and biomaterials have evolved significantly, achieving widespread clinical applications across fields, such as orthopedics, dentistry, plastic/reconstructive surgery, and cardiovascular medicine[132].

Early commercial medical products, such as Biobrane®, utilized purified ECM proteins as coating materials. With the emergence and advancement of decellularization technologies, tissue-derived ECM has gained prominence as a scaffold material in clinical practice due to its ability to retain most protein components and native 3D structure. A representative example is BioCartilage® (developed by Arthrex), an acellular cartilage matrix product derived from allogeneic cartilage. It preserves critical matrix components like type II collagen and serves as an adjunctive scaffold for microfracture surgery. Beyond providing structural support, it enhances repair outcomes for articular cartilage defects by facilitating autologous cell migration and functional regeneration through bioactive signaling. Another successful case is TissueMend™, a collagen matrix membrane approved by the Food and Drug Administration and Conformité Européenne. Processed from decellularized fetal bovine dermis, it is widely used in tendon repair, leveraging its unique biocompatibility and cell-regulatory capabilities to promote tissue regeneration and suppress inflammatory responses[133].

Compared to tissue/organ-derived ECM, mECM offers advantages, such as avoiding donor immunogenicity and enhancing differentiation

potential, holding immense promise for clinical tissue engineering and cell therapies[37]. Current research focuses on integrating mECM with natural or synthetic polymers, ceramics, and metals to engineer advanced biomaterials —including hydrogels, electrospun fibers, and 3D-printed constructs for applications in bone, cartilage, tendon, and ligament regeneration. Preclinical studies have demonstrated clear therapeutic efficacy.

However, the clinical translation of mECM faces significant challenges: biological heterogeneity leading to batch variability, quality control in scaled production, validation of efficacy for specific indications, regulatory classification standards, and cost constraints. Breakthroughs in these critical areas will determine the ultimate translational value of mECM products in regenerative medicine.

## Conclusions and outlooks

Within the domain of musculoskeletal tissue regeneration, mECM has been extensively researched and applied as a cell culture substrate, material coating, scaffold material, and hydrogel component. Compared to dECM from other sources, mECM has significant unique advantages. Firstly, mECM can be prepared using the patient's own MSCs, which greatly reduces the risk of immune rejection after transplantation and avoids the potential pathogen transmission issues associated with allogeneic or xenogeneic cells. Moreover, based on stem cells with strong bioactivity, a large amount of high-quality mECM can be produced, providing a more reliable source of biomaterials for tissue engineering and regenerative medicine. Most importantly, mECM has a high degree of controllability and customization potential. Through cell culture systems, researchers can precisely regulate the composition, structure, and bioactivity of mECM, providing ideal biomaterials for various biological research and clinical applications.

However, in the process of designing and manufacturing mECM materials that meet the complex needs of musculoskeletal tissue repair, there are still many challenges, including

(i) The lack of standardized decellularization guidelines. Different studies follow flexible decellularization methods, making it difficult to determine which method is best suited for specific applications. Establishing widely accepted decellularization evaluation standards among researchers can help compare different decellularization methods;

(ii) The mechanism by which mECM regulates cell behavior has not yet been fully elucidated. Both autologous and allogeneic mECM can effectively promote cell growth and have various cell behavior regulatory effects, but the understanding of the specific molecular mechanisms and the main matrix components involved in these mechanisms is very limited at present. By integrating the latest proteomics and bioinformatics technologies, there will be an opportunity to clarify the complex mechanisms of mECM regulation of cell behavior;

(iii) mECM from different types of stem cells has a unique structure and composition, and even mECM from the same type but from different differentiation stages of stem cells also shows significant differences, which will importantly impact cell behavior and function. Research and identification of the best stem cell source for mECM to meet specific tissue repair purposes will help prepare more targeted mECM-derived biomaterials;

(iv) The production cost of mECM is relatively high, and it is necessary to explore cost-effective production methods for future widespread clinical treatment;

(v) For clinical applications, it is vital to develop methods for large-scale production of high-quality and consistent mECM;

(vi) Further studies are required to evaluate the long-term stability and biosafety of mECM in vivo, including its potential to induce long-term immune reactions or impact tissue function;

(vii) Although mECM provides a biocompatible microenvironment conducive to cell adhesion, migration, and differentiation, its own mechanical strength and stability may not be sufficient to support some tissue engineering applications, especially in tissues that need to withstand greater mechanical loads (such as bones, tendons, and

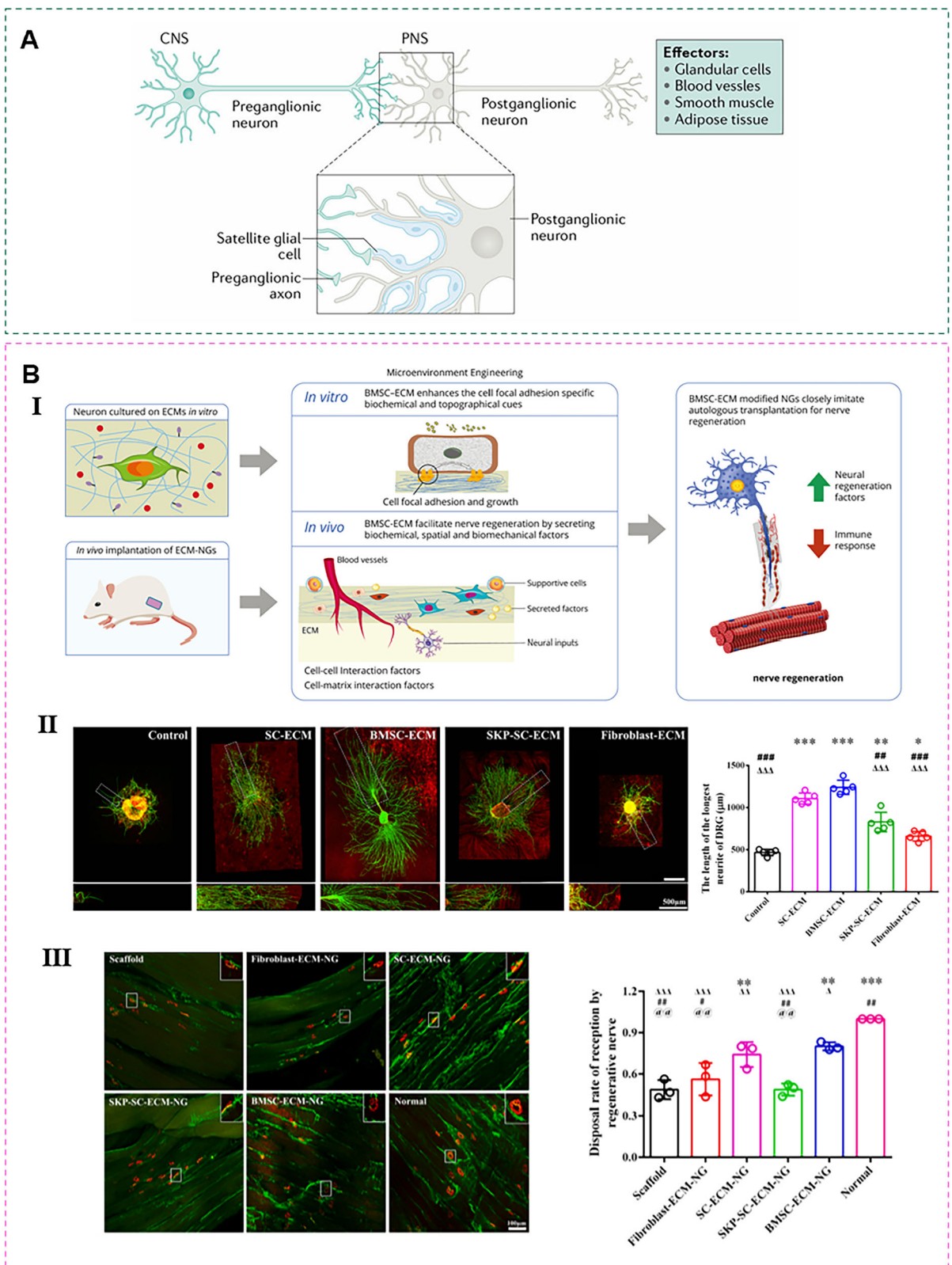

**Fig. 6 | Application of mECM in nerve. A** Neuron hierarchical structure[142]. **B** mECM promotes axonal growth in the rat sciatic nerve and reinnervation of the muscles (tibialis anterior and gastrocnemius). **I** A schematic illustrating the mechanisms through which mECM promotes nerve regeneration. **II** BM-MSCs-ECM enhanced the axonal outgrowth in the dorsal root ganglions (DRGs). **III** BMSC-ECM-NG promotes nerve regeneration and further myelin regeneration[123].

cartilage). Once these lingering challenges can be further detailed and resolved, they are likely to become an innovative paradigm for future clinical applications.

In summary, despite the aforementioned drawbacks, mECM is still considered one of the ideal materials for future musculoskeletal tissue regeneration and repair due to its high biocompatibility and customizability. The use of autologous mECM can effectively avoid the risk of immune rejection, and its properties can be adjusted through in vitro culture conditions to achieve personalized customization for specific biological functions and clinical needs. This offers a promising inspiration for the design of future biomaterials and is an optimized solution for problems in tissue engineering and regenerative therapy. However, due to the complex composition and structure of ECM, we urgently need newer proteomic techniques to capture the diversity of ECM protein forms and construct spatially resolved ECM maps[134].

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

## Acknowledgements

This study was supported by the National Natural Science Foundation of China (82302664,82572842,82502907), the Basic Research Program of Jiangsu Province (BK20230494, BK20250833) and Gusu Innovation and Entrepreneur Leading Talents project (ZXL2023204), National High-level Young Talent Program (KS24700124), Jiangsu Specially Appointed Professor Program (SR24700123), Priority Academic Program Development of Jiangsu Higher Education Institutions (PAPD), the Project of MOE Key Laboratory of Geriatric Diseases and Immunology (no. JYN202504), the Foundation of National Center for Translational Medicine (Shanghai) SHU Branch (UITM.202501), the Postdoctoral Fellowship Program (Grade C) of China Postdoctoral Science Foundation (GZC20251567), and Jiangsu Funding Program for Excellent Postdoctoral Talent (2025ZB128), the China Postdoctoral Science Foundation (2024M762305), the Wuxi "Light of Taihu" Medical Science and Technology Project (Y20242210). Figures 1 and 2 were generated using icons and other elements from Biorender.

## Author contributions

All authors were involved in drafting the article or revising it critically for important intellectual content, and all authors approved the final version to be published. Supervision and conceptualization: G.Z., X.Z., and Y.X. Investigation: S.L., J.W., and J.C. Editing: Y.Y. and X.H.

## Competing interests

The authors declare no competing interests.
