## [Transparent Peer Review File · Communications Biology]

Mesenchymal Stem Cell-Derived Extracellular Matrix for Musculoskeletal Tissue Regeneration

Corresponding Author: Professor Yong Xu

Version 1:

Reviewer comments:

Reviewer #1

(Remarks to the Author)

The manuscript entitled "Mesenchymal Stem Cell-Derived Extracellular Matrix for Musculoskeletal Tissue Regeneration" addresses a highly relevant and timely topic in the field of regenerative medicine and tissue engineering. The limited self-healing capacity of tissues such as cartilage, tendon, bone, and other components of the musculoskeletal system, combined with the lack of effective tissue grafts and biomaterials for repairing damaged tissue, highlights the urgent need to explore new sources of highly biocompatible materials capable of stimulating tissue regeneration.

This review focuses on the use of mesenchymal stem cell-derived extracellular matrix (mECM) as a cell-free bioactive scaffold that can recapitulate the native microenvironment, providing important insights into the biological properties, fabrication techniques, and both in vitro and in vivo studies of mECM efficacy.

The review includes key studies in the field and provides a critical evaluation by the authors. The structure is generally logical, but it could be improved. Please see the recommendations below.

Major Comments

1. In sub-chapter 2.1. "The Structure and Components of mECM" the description of mECM is rather superficial. It is recommended to provide more detailed information, and to include a discussion of structural differences compared with tissue-derived ECM.
2. In sub-chapter 2.2. "Decellularizing the mECM," page 5, lines 98–101, the phrase "DNA and the α -galactoside (α -Gal) epitope on cell surfaces are considered major antigens" requires clarification. It should be explained that α -Gal is typically found in xenogeneic non-primate tissues; otherwise, the statement is unclear, since the discussion also includes allografts.
3. In sub-chapter 2.2. "Decellularizing the mECM," it is recommended to include a summary table of studies that have employed different decellularization methods, as well as their combinations, to obtain mECM.
4. In sub-chapter 2.3. "Characteristic Differences of mECM from Different Sources of MSCs," page 6, lines 128–130, after the sentence "Although ECM from different stem cell sources shares a subset of common proteins, their composition is markedly distinct from that of ECM derived from somatic cells, such as fibroblasts", the specific differences in protein composition should be provided. In the examples mentioned afterward, these differences are not clearly presented.
5. Page 13, line 249. The phrase "mECM sheets produced from osteogenic cell sheets" is unclear, particularly regarding what the authors mean by "osteogenic cell sheet." This should be explained in the text.
6. Page 15, line 266: The authors state, "When mECM-pretreated SM-MSCs were cultured as particles or injected into cartilage defects..." It should be clarified what "pretreated" means, and it should also be specified that the cells were not only "cultured as particles," but that their differentiation ability in vitro was studied.
7. Page 15, line 286. The terms "mECM mimicking early chondrogenesis" and "mECM mimicking late chondrogenesis" should be explained. Using these terms outside the context of the original referenced paper, without clarification, may cause confusion for readers.

8. In sub-chapter 4.3. "The application of mECM in muscle regeneration", page 20, lines 241-247, the discussion on studies specifically focused on mECM is rather brief, while most of the content centers on the use of tissue-derived ECM. It is recommended to expand this section by providing more detailed information on the application of mECM in muscle regeneration.

9. It is recommended to move Chapter 6. "The mechanism of mECM affecting cell behaviors" to follow Chapter 3. "The effect of mECM on cell behavior" since both chapters discuss the effects of mECM on cells.

10. Page 27, line 475-478. When discussing this study, the type of cells should be mentioned.

Minor Comments

1. Page 3, line 46. The statement that the viscoelasticity of ECM decreases with age should be supported by an appropriate reference.

2. Page 3, lines 62–63. What stem cells are meant by "intact stem cells"? Are the authors referring to MSCs?

3. Page 3, lines 64. A period is missing after the word "advantages".

4. Page 7, line 152. Which stem cells are meant? This should be clarified.

5. Page 8, line 159-160. "MSCs-ECM" should be corrected to "mECM".

6. Page 10, line 174. The word "that" should be removed.

7. Page 10, line 181. Which stem cells are meant? This should be clarified.

8. Page 13, line 234. It is unclear what "MSCs pretreated with mECM" means. Were they cultured on mECM? This should be clarified.

9. Page 15, line 278. The abbreviation "OA" should be defined.

10. Page 16, line 281. What kind of "stem cells" are being referred to? This should be clarified.

11. Page 19, line 321. Which stem cells are meant? Are these satellite cells? This should be specified.

12. Page 21, lines 376-377. A reference should be added.

13. Page 22, lines 388-391. A reference should be added.

14. Pages 22-23, lines 392-413. This section of text lacks references and should be properly cited.

15. Page 23, line 421. The term "ECM-SPS scaffolds" should be explained.

16. Page 24, line 425. The abbreviations "COMP" and "TSP-1" should be defined.

17. Pages 26, lines 450-458. This section should be properly cited.

18. Pages 26, lines 462-467. This section should be properly cited.

19. Page 27, line 476. The abbreviation "COL I" and "TSP-1" should be defined.

20. Table 1 should be revised, as the lines in the columns appear to be misaligned.

Thus, the manuscript addresses a timely and highly important topic in regenerative medicine. The use of mECM as a bioactive scaffold has great potential to advance musculoskeletal tissue repair, making this review both relevant and impactful for the field. However, despite its strengths, the manuscript requires substantial improvements in structure, depth of discussion, clarity of terminology, and citation support.

Reviewer #2

(Remarks to the Author)

The manuscript entitled "Mesenchymal Stem Cell-Derived Extracellular Matrix for Musculoskeletal Tissue Regeneration" provides a review of Mesenchymal Stem Cell-Derived Extracellular Matrix. The topic is of potential interest to readers in the field of Tissue engineering. The review covers a number of important aspects, but in its current form the manuscript needs revisions to improve. With revision, this paper could be a valuable resource.

1. When the authors talk about the mECM acquisition methods, they mentioned non-ionic detergents or chelating agents. However, in the following text, they only give a description of non-ionic detergents. The introduction of method using

chelating agents should be included before giving the optimization of methods.

2. The use of Figure in the manuscript is insufficient. The authors should clearly indicate the location of each figure, including the location of individual figures within the figure gallery. For example, Figure 1a should appear on page 5, and Figure 1b should appear on page 3 or 4. It is hard to track the figures in the current version. Furthermore, the location of Figure 4 is missing in the manuscript.

3. I noticed that there are some tables in the supporting information, but the author did not give any description in the manuscript. If the authors want to use these tables, they should be included in the manuscript as Table S1, S2... and describe them in detail.

4. Figures and tables about applications in muscle and tendon/ligaments should also be included. In Figure 5, figures about blood vessels should also be included. Besides, if the authors want to include the tendon/ligaments in the manuscript, they should also mention it in the abstract and introduction sections.

5. The English writing in the whole manuscript should be carefully checked, especially the use of singular and plural forms. Grammatical errors should be carefully corrected.

Version 2:

Reviewer comments:

Reviewer #1

(Remarks to the Author)

I thank the authors for addressing my questions and incorporating the necessary corrections into the text. At this stage, I have only a couple of comments:

1. In sub-chapter 2.2 "Decellularizing the mECM", it appears that the authors omitted the description of chelating agents (provided in response to Reviewer #2's comment), which was mentioned in the Rebuttal Letter. Please make sure this is included in the main text.

2. There are two tables labeled as Table S2 in the SUPPLEMENTAL MATERIAL. Kindly revise the table numbering and verify that all references to these tables in the main text are accurate.

Dear Editor,

We would like to express our gratitude for the thorough review of our manuscript entitled "Mesenchymal Stem Cell-Derived Extracellular Matrix for Musculoskeletal Tissue Regeneration (COMMSBIO-25-5836B)". We sincerely appreciate the valuable and constructive feedback provided by the reviewers. In response to the reviewers' recommendations, we have diligently revised the manuscript, incorporating the suggested changes and outlining additional references. The specific comments provided by the reviewers have been addressed comprehensively. We believe that the revised version of the manuscript now aligns with the high standards set forth by the *Communications Biology* Letters for publication. We thank the editorial team and reviewers for their time and consideration.

Best Regards,

Yong Xu, Ph.D. Professor

Orthopaedic Institute, Soochow University

Email address: yxu1615@suda.edu.cn

Reviewer #1 (Remarks to the Author): The manuscript entitled "Mesenchymal Stem Cell-Derived Extracellular Matrix for Musculoskeletal Tissue Regeneration" addresses a highly relevant and timely topic in the field of regenerative medicine and tissue engineering. The limited self-healing capacity of tissues such as cartilage, tendon, bone, and other components of the musculoskeletal system, combined with the lack of effective tissue grafts and biomaterials for repairing damaged tissue, highlights the urgent need to explore new sources of highly biocompatible materials capable of stimulating tissue regeneration. This review focuses on the use of mesenchymal stem cell-derived extracellular matrix (mECM) as a cell-free bioactive scaffold that can recapitulate the native microenvironment, providing important insights into the biological properties, fabrication techniques, and both in vitro and in vivo studies of mECM efficacy. The review includes key studies in the field and provides a critical evaluation by the authors. The structure is generally logical, but it could be improved. Please see the recommendations below.

P
A
G
E

\

Major Comments

1. In sub-chapter 2.1. “The Structure and Components of mECM” the description of mECM is rather superficial. It is recommended to provide more detailed information, and to include a discussion of structural differences compared with tissue-derived ECM.

Authors’ response: Thank you very much for your valuable suggestion. We have thoroughly revised this section: mECM is a 3D matrix supporting cell attachment, migration, and function—under SEM, it presents randomly arranged nanofiber bundles with a porous structure that facilitates nutrient diffusion, oxygen transport, and cell infiltration, while AFM shows it has tissue-specific mechanical properties (Young’s modulus 0.1-10 kPa; e.g., bone marrow-derived mECM is stiffer due to dense collagen packing, adapting to the mandibular microenvironment) and regulates cell behavior via mechanotransduction; compositionally, it mainly consists of collagen (type I forms a fibrillar backbone for tensile strength, type IV assembles into basement membrane-like domains), fibronectin (with RGD sequences binding integrin $\alpha 5\beta 1$ to activate FAK signaling for cell migration/survival), proteoglycans (e.g., decorin regulating collagen assembly), and hyaluronic acid (maintaining matrix hydration via CD44 interactions to enhance regenerative capacity). We also added a comparison with tissue-derived ECM (tECM): unlike mECM’s uniform nanofiber structure, controllable porosity, and stable tissue-specific mechanics, tECM has heterogeneous fiber diameters, uneven pores, and drastically variable stiffness due to donor/tissue differences, and its harsh decellularization may disrupt bioactive components—these differences highlight mECM’s superiority for supporting ordered cell aggregate formation and mandibular regeneration. These revisions are integrated into Sub-chapter 2.1 to enhance depth and clarity.

Changes Made:

mECM is a three-dimensional (3D) matrix that supports cell attachment, migration, survival, and function. Under scanning electron microscopy (SEM), mECM appears as randomly arranged nanofiber bundles¹. This porous architecture is essential for facilitating nutrient diffusion, oxygen transport, and waste exchange, while also providing sufficient space for cell infiltration during tissue regeneration. Atomic force microscopy (AFM)

P
A
G
E

\
*

studies further reveal that mECM exhibits tissue-specific mechanical properties, with Young's modulus varying from 0.1 to 10 kPa²; for instance, bone marrow-derived mECM is relatively stiffer due to dense collagen packing, whereas adipose-derived mECM is softer to match the mechanical microenvironment of soft tissues. Such structural features enable mECM to recapitulate the native tissue microenvironment and regulate cell behavior through mechanotransduction.

Studies have shown that mECM is primarily composed of collagen, fibronectin (FN), laminin, elastin, and other adhesive proteins, as well as various proteoglycans and hyaluronic acid³. Among collagens, type I collagen forms the fibrillar backbone to confer tensile strength, while type IV collagen assembles into sheet-like networks that constitute basement membrane-like domains, supporting cell polarization and adhesion⁴. Fibronectin contains conserved RGD (Arg-Gly-Asp) sequences that specifically bind to cell surface integrins (e.g., $\alpha 5 \beta 1$), activating focal adhesion kinase (FAK)-mediated signaling pathways to promote cell migration and survival⁵. Proteoglycans such as decorin regulate collagen fibril assembly and matrix integrity by enhancing molecular adhesion between aggrecan and collagen⁶, while aggrecan forms large complexes with hyaluronic acid to endow mECM with compressive resilience. Hyaluronic acid, a key glycosaminoglycan, maintains matrix hydration and mediates cell proliferation via interactions with the CD44 receptor⁷, which partly explains the elevated pro-regenerative capacity of young mECM with higher hyaluronic acid content⁸.

When compared to tissue-derived decellularized ECM (tECM), mECM presents distinct structural and compositional characteristics, with each offering unique potential advantages. tECM retains the complex, hierarchical architecture and tissue-specific bio-molecular composition of the native organ, providing a holistic microenvironment that can be difficult to fully replicate *in vitro*⁹. It is important to note that advancements in decellularization technologies, such as the use of supercritical CO₂ and mild bio-detergents, have significantly

improved the preservation of crucial ECM components while reducing immunogenic residues in tECM scaffolds¹⁰. The inherent complexity of tECM, which mirrors the *in vivo* niche, may indeed be critical for orchestrating complex regenerative processes.

In contrast, mECM lacks the macroscopic, organ-specific tissue hierarchy of tECM. Its structure is more homogeneous and originates from a single, defined cell population. This defined origin is the source of its potential advantages, including a minimized risk of immunogenicity (particularly for autologous applications) and a high degree of controllability and customizability. The composition, stiffness, and bioactivity of mECM can be tailored by pre-conditioning the MSCs during the deposition phase. However, whether this tailored homogeneity is superior to the inherent complexity of tECM for clinical outcomes lacks long-term translational data. Ultimately, the choice between mECM and tECM may be application-dependent, revolving around the specific trade-off between a highly complex, native microenvironment (tECM) and a well-defined, tunable, and potentially patient-specific platform (mECM).

- In sub-chapter 2.2. “Decellularizing the mECM,” page 5, lines 98–101, the phrase “DNA and the α -galactoside (α -Gal) epitope on cell surfaces are considered major antigens” requires clarification. It should be explained that α Gal is typically found in xenogeneic non-primate tissues; otherwise, the statement is unclear, since the discussion also includes allografts.*

Authors’ response: We sincerely thank the reviewer for this insightful comment and for pointing out the need for greater precision in our discussion of immunogenic antigens. The reviewer is absolutely correct that the α -galactoside (α -Gal) epitope is predominantly a concern in xenogeneic tissues (e.g., from pigs or bovines), whereas it is not present in

P
A
G
E

\

human allografts. Our original statement was indeed unclear in this regard.

As suggested, we have revised the manuscript to clarify this critical distinction. The text now explicitly states that **DNA is a primary concern for both allogeneic and xenogeneic grafts, while the α -Gal epitope is a major immunogen specific to xenogeneic sources.** We believe this modification significantly improves the accuracy and clarity of this section.

3. *In sub-chapter 2.2. “Decellularizing the mECM,” it is recommended to include a summary table of studies that have employed different decellularization methods, as well as their*

Method Category	Specific Protocol	MSC Source	Key Findings / Outcome
Biological/Chemical	0.5% SDS+1%Triton X-100 + DNase	BM-MSCs	It has good biocompatibility, is non-cytotoxic, and can promote the osteogenic and chondrogenic differentiation of BM-MSCs ¹¹ . Effective DNA removal; preserved glycosaminoglycan (GAG) ¹²
Physical/Chemical	cryogenic milling + freeze-drying + dehydrothermal	BM-MSCs	content and nano-fibrous structure; promoted chondrogenic differentiation.
Physical/Chemical	Freeze-Thaw cycles + 25 mM NH ₄ OH	MSCs	Nucleus was completely removed, with extremely low residual DNA (4.38 ± 1.67 ng), and the ECM structure was intact ¹³ .
Chemical	0.1% Triton X-100 + 1.5M KCl	MSCs	Nucleus was completely removed, with extremely low residual DNA (7.24 ± 2.15 ng), and the ECM structure was intact ¹³ .
Biological/Chemical	0.5% Triton X-100 + 20 mM NH ₄ OH + DNase	UC-MSCs	No cell nucleus residue, promoting the survival of neurons and axon growth ¹⁴ .

combinations, to obtain mECM.

Authors’ response: We thank the reviewer for this excellent suggestion. We agree that a summary table will greatly enhance the clarity and utility of the decellularization section. As requested, we have included a new table summarizing key studies that have employed various methods to decellularize mECM.

Table 1 different decellularization methods

4. *In sub-chapter 2.3. “Characteristic Differences of mECM from Different Sources of MSCs,” page 6, lines 128 130, after the sentence “Although ECM from different stem cell sources shares a subset of common proteins, their composition is markedly distinct from that of ECM derived from somatic cells, such as fibroblasts”, the specific differences in protein composition should be provided. In the examples mentioned afterward, these differences*

are not clearly presented.

Authors' response: We thank the reviewer for this excellent suggestion. Cells from different tissue sources, including BM-MSCs, AD-MSCs, and neonatal dermal fibroblasts (NHDF), although the ECM deposited *in vitro* shares a core matrix histone (such as collagen I, III, VI, fibronectin), The common basis of tendinin C, etc., but its unique "matrix group characteristics" have been revealed through quantitative proteomics analysis. These differences are reflected in the composition and abundance of specific proteins: **For example, BM-MSCs ECM is rich in factors related to the bone marrow microenvironment (such as CXCL12 and S100 proteins); AD-MSCs ECM specifically expresses tendinin XB (TNXB) and connective tissue growth factor (CTGF). The overall collagen content of Der ECM is relatively low, but it is rich in fibrinogen 2 (FBN2) and proteins related to the TGF- β and WNT signaling pathways (such as LTBP4, WNT5A). These specific protein composition characteristics reflect the functional specificity of their original tissues and differentially regulate the transcriptome and behavior of cells, emphasizing the importance of considering ECM sources in tissue engineering for precisely simulating the *in vivo* microenvironment.**

5. *Page 13, line 249. The phrase "mECM sheets produced from osteogenic cell sheets" is unclear, particularly regarding what the authors mean by "osteogenic cell sheet." This should be explained in the text.*

Authors' response: We thank the reviewer for this comment. The term "osteogenic cell sheet" (OCS) refers to a specific construct in tissue engineering, fabricated from BM-MSCs, which becomes enriched with osteoblasts, ECM, and osteogenic growth factors through the culture process¹⁵. As the reviewer rightly points out, this should be clarified in the text. We have now added a brief explanation and relevant references to the manuscript at the first mention of OCS to ensure clarity for all readers.

We have amended the text on **Page 18, line 328**, accordingly to incorporate these clarifications.

6. *Page 15, line 266: The authors state, “When mECM-pretreated SM-MSCs were cultured as particles or injected into cartilage defects...” It should be clarified what “pretreated” means, and it should also be specified that the cells were not only “cultured as particles,” but that their differentiation ability in vitro was studied.*

Authors’ response: We appreciate the reviewer's insightful comment regarding the need for greater precision in our methodology description.

Regarding "pretreated," we have clarified that it specifically means the SM-MSCs were expanded and cultured on the mECM substrate prior to subsequent experiments. This pretreatment step leverages the mECM's ability to enhance the cells' chondrogenic potential. We have amended the text on **Page 18, line 312**, accordingly to incorporate these clarifications.

7. *Page 15, line 286. The terms “mECM mimicking early chondrogenesis” and “mECM mimicking late chondrogenesis” should be explained. Using these terms outside the context of the original referenced paper, without clarification, may cause confusion for readers.*

Authors’ response: We thank the reviewer for this valuable comment. We agree that the terms "mECM mimicking early chondrogenesis" and "mECM mimicking late chondrogenesis" require clarification for readers unfamiliar with the specific context of the reference (Cai et al.)¹⁶.

As defined in the cited study, these terms refer to mECM deposited by MSCs that were cultured in chondrogenic induction medium for different durations. Specifically:

"mECM mimicking early chondrogenesis" is derived from MSCs after a relatively short period (e.g., 1 week) of chondrogenic induction, resulting in a matrix that begins to accumulate cartilage-specific components.

"mECM mimicking late chondrogenesis" is derived from MSCs after a longer period (e.g., 3 weeks) of chondrogenic induction, resulting in a matrix rich in mature cartilage-specific components.

We have revised the manuscript on **Page 21, line 367-370**, to include a brief explanation of these terms, ensuring clarity for all readers.

8. *In sub-chapter 4.3. “The application of mECM in muscle regeneration”, page 20, lines 241-247, the discussion on studies specifically focused on mECM is rather brief, while most of the content centers on the use of tissue derived ECM. It is recommended to expand this section by providing more detailed information on the application of mECM in muscle regeneration.*

Authors’ response: We sincerely thank the reviewer for this insightful and constructive comment. We agree that the original version did not sufficiently highlight the unique applications and potential of mesenchymal stem cell-derived ECM (mECM) in muscle regeneration. As suggested, we have extensively revised and expanded Section 4.3 to provide a more detailed and focused discussion on mECM.

Changes Made:

Although various decellularized matrices have been utilized for skeletal muscle regeneration, most current studies still adopt a "cell-matrix co-delivery" strategy, where cells and matrices are implanted together into the injury site, without achieving standardized, independent separation and application of ECM¹⁷. Although this approach promotes functional recovery to some extent, it remains difficult to clearly distinguish whether the effects are attributable to the cells or the matrix, thereby limiting the systematic evaluation and standardized preparation of ECM as an independent biomaterial. Wang et al demonstrated that injecting decellularized adipose-derived stem cell ECM fragments as a bulking agent into the urethra of a stress urinary incontinence (SUI) rat model not only achieved physical bulking but, more importantly, recruited host cells and promoted local smooth muscle regeneration, significantly restoring urethral function. This highlights the great potential of ADSC-ECM in functional muscle regeneration¹⁸.

9. *It is recommended to move Chapter 6. “The mechanism of mECM affecting cell behaviors” to follow Chapter 3. “The effect of mECM on cell behavior” since both chapters discuss the effects of mECM on cells.*

Authors’ response: We thank the reviewer for this excellent suggestion. We agree that

P
A
G
E

\

placing the "Mechanism" chapter immediately after the "Cell Behavior" chapter significantly improves the logical flow of the manuscript. This rearrangement allows readers to first understand the phenotypic effects of mECM on cells and then immediately delve into the underlying molecular mechanisms, creating a more coherent and intuitive narrative. We have accordingly moved Chapter 6 to follow Chapter 3.

10. Page 27, line 475-478. When discussing this study, the type of cells should be mentioned.

Authors' response: We thank the reviewer for pointing out this omission. We have revised the text on **Page 13, lines 241**, to explicitly mention the specific type of mesenchymal stem cells (human umbilical cord mesenchymal stem cells) used in the discussed study, as per the original reference. This clarification enhances the accuracy of our description.

Minor Comments

1. Page 3, line 46. The statement that the viscoelasticity of ECM decreases with age should be supported by an appropriate reference.

Authors' response: We thank the reviewer for this suggestion. We agree that the statement regarding age-related changes in ECM viscoelasticity requires supporting evidence. We have now cited the review by Selman & Pardo (2021), which comprehensively discusses the increase in ECM stiffness due to crosslinking during ageing, at the appropriate location (**Page 3, line 48**) to robustly support this claim¹⁹.

2. Page 3, lines 62–63. What stem cells are meant by "intact stem cells"? Are the authors referring to MSCs?

Authors' response: We thank the reviewer for this astute observation. The term "intact stem cells" was indeed intended to refer to functional, unmodified mesenchymal stem cells (MSCs), in contrast to the acellular mECM material derived from them. To avoid any confusion, we have revised the text on **Page 3, lines 65**, to explicitly state "mesenchymal stem cells (MSCs)" instead of the ambiguous "intact stem cells."

3. *Page 3, lines 64. A period is missing after the word “advantages”.*

Authors’ response: We thank the reviewer for their careful reading. The missing period after "advantages" on **Page 4, line 67**, has been added.

4. *Page7, line 152. Which stem cells are meant? This should be clarified.*

Authors’ response: We thank the reviewer for pointing out the need for clarification. The term "stem cells" in this context refers specifically to synovial membrane-derived mesenchymal stem cells (SM-MSCs). We have revised the text on **Page 10, line 195**, accordingly to avoid any ambiguity.

5. *Page 8, line 159-160. “MSCs-ECM” should be corrected to “mECM”.*

Authors’ response: We sincerely thank the reviewer for pointing out this inconsistency in terminology. The correction has been made as suggested.

6. *Page 10, line 174. The word “that” should be removed.*

Authors’ response: We thank the reviewer for their careful reading and for identifying this grammatical error. The suggested change has been made.

7. *Page10, line 181. Which stem cells are meant? This should be clarified.*

Authors’ response: We thank the reviewer for this query. The term "stem cells" in this sentence refers to the bone marrow mesenchymal stem cells (BM-MSCs). The sentence has been clarified accordingly. We have revised the text on **Page 12, line 224**, accordingly to avoid any ambiguity.

8. *Page 13, line 234. It is unclear what “MSCs pretreated with mECM” means. Were they cultured on mECM? This should be clarified.*

Authors’ response: We apologize for the lack of clarity. The reviewer is correct. "MSCs pretreated with mECM" means that the MSCs were cultured on the mECM substrate during the expansion phase prior to their transplantation or further experimentation. The text has

been revised to state this explicitly. We have revised the text on **Page 18, line 328**, accordingly to avoid any ambiguity.

9. *Page 15, line 278. The abbreviation "OA" should be defined.*

Authors' response: We thank the reviewer for pointing this out. The abbreviation **osteoarthritis** "OA" has been defined upon its first use in the manuscript.

10. *Page 16, line 281. What kind of "stem cells" are being referred to? This should be clarified.*

Authors' response: We apologize for the ambiguity. In this context, the "stem cells" refer specifically to the adipose stem cells that were the subject of the cited study. The text has been clarified. We have revised the text on **Page 21, line 362**, accordingly to avoid any ambiguity.

11. *Page 19, line 321. Which stem cells are meant? Are these satellite cells? This should be specified.*

Authors' response: The reviewer is correct to seek clarification. In this sentence, the "stem cells" refer to the tissue-resident muscle stem cells, which are indeed the **satellite cells**. This has been specified in the revised text.

12. *Page 21, lines 376-377. A reference should be added.*

Authors' response: We are grateful to the reviewer for this valuable feedback. In response, we have cited relevant literature to strengthen our claims.

13. *Page 22, lines 388-391. A reference should be added.*

Authors' response: We are grateful to the reviewer for this valuable feedback. In response, we have cited relevant literature to strengthen our claims.

14. *Pages 22-23, lines 392-413. This section of text lacks references and should be properly*

cited.

Authors' response: We agree with the reviewer and apologize for the oversight. Appropriate citations have been added to the specified section (Pages 28) to support the statements regarding nerve regeneration and the mechanical properties of the composite scaffolds.

15. Page 23, line 421. The term "ECM-SPS scaffolds" should be explained.

Authors' response: We thank the reviewer for suggesting this clarification. The term "ECM-SPS scaffolds" has been explained in detail, drawing upon the context of the cited source, to clearly convey its composition and functional rationale.

Extracellular matrix-synthetic polymer scaffold (ECM-SPS) design creates a biomimetic bilayer scaffold, where the synthetic polymers provide the requisite mechanical strength and structural integrity, while the tissue-specific ECM components replicate the native biochemical microenvironment to guide cell behavior and tissue regeneration.

16. Page 24, line 425. The abbreviations "COMP" and "TSP-1" should be defined.

Authors' response: We thank the reviewer for pointing this out. The abbreviations "COMP (Cartilage oligomeric matrix protein)" and "TSP-1 (thrombospondin 1)" have been defined upon their first use in the manuscript.

17. Pages 26, lines 450-458. This section should be properly cited.

Authors' response: We are grateful to the reviewer for this valuable feedback. In response, we have cited relevant literature to strengthen our claims.

Arthrex,

BioCartilage®.

<https://www.arthrex.com/resources/presentation/fJ3kA93jWki3BQE9fyQCXw/biocartilag>
e, 2013.

18. Pages 26, lines 462-467. This section should be properly cited.

Authors' response: We are grateful to the reviewer for this valuable feedback. In response,

P
A
G
E

\

we have cited relevant literature to strengthen our claims.

19. Page 27, line 476. The abbreviation “COL I” and “TSP-1” should be defined.

Authors’ response: We apologize for the oversight. The abbreviations "COL I (collagen type I)" and "TSP-1(thrombospondin 1)" have been defined upon their first use in the specified sentence.

20. Table 1 should be revised, as the lines in the columns appear to be misaligned.

Authors’ response: We thank the reviewer for noting this formatting issue. Table 1 has been revised to ensure proper alignment of the columns for improved readability.

Thus, the manuscript addresses a timely and highly important topic in regenerative medicine. The use of mECM as a bioactive scaffold has great potential to advance musculoskeletal tissue repair, making this review both relevant and impactful for the field. However, despite its strengths, the manuscript requires substantial improvements in structure, depth of discussion, clarity of terminology, and citation support.

Reviewer #2 (Remarks to the Author): *The manuscript entitled “Mesenchymal Stem Cell-Derived Extracellular Matrix for Musculoskeletal Tissue Regeneration” provides a review of Mesenchymal Stem Cell-Derived Extracellular Matrix. The topic is of potential interest to readers in the field of Tissue engineering. The review covers a number of important aspects, but in its current form the manuscript needs revisions to improve. With revision, this paper could be a valuable resource.*

1. *When the authors talk about the mECM acquisition methods, they mentioned non-ionic detergents or chelating agents. However, in the following text, they only give a description of non-ionic detergents. The introduction of method using chelating agents should be included before giving the optimization of methods.*

Authors’ response: Thank you for your precise comment on the incomplete introduction of mECM decellularization methods. We have revised the content to supplement the description of chelating agent-based decellularization before discussing method optimization, as follows:

P
A
G
E

\

Chelating agents, represented by ethylenediaminetetraacetic acid (EDTA) and ethylene glycol tetraacetic acid (EGTA), function by sequestering divalent cations (e.g., Ca^{2+} , Mg^{2+}) critical for maintaining cell membrane integrity and intercellular junctions¹³. This cation depletion disrupts the structural stability of cell membranes, leading to gradual cell lysis, while their mild chemical nature minimizes damage to sensitive ECM components (e.g., collagen triple helices, glycosaminoglycan chains) that are vital for mECM's biomimetic activity²⁰. For instance, EDTA is frequently used to loosen cell adhesion to the ECM scaffold by chelating Ca^{2+} in focal adhesion complexes, and it is often combined with nucleases (DNase/RNase) to further eliminate residual genomic DNA and RNA—similar to the synergistic use of non-ionic detergents with nucleases. Notably, EGTA exhibits higher selectivity for Ca^{2+} over Mg^{2+} , making it preferable when preserving Mg^{2+} -dependent ECM enzymes (e.g., lysyl oxidase) is required for subsequent tissue regeneration²¹.

This revision ensures a complete and logical presentation of the two main decellularization approaches, which has been integrated into the corresponding section of the manuscript.

2. The use of Figure in the manuscript is insufficient. The authors should clearly indicate the location of each figure, including the location of individual figures within the figure gallery. For example, Figure 1a should appear on page 5, and Figure 1b should appear on page 3 or 4. It is hard to track the figures in the current version. Furthermore, the location of Figure 4 is missing in the manuscript.

Authors' response: We sincerely thank the reviewer for this critical feedback regarding the presentation and placement of figures within our manuscript. We acknowledge that the original submission made it difficult to track the figures and their corresponding discussions in the text. We have thoroughly revised the manuscript to address this issue.

3. I noticed that there are some tables in the supporting information, but the author did not give any description in the manuscript. If the authors want to use these tables, they should be included in the manuscript as Table S1, S2... and describe them in detail.

Authors' response: We thank the reviewer for their attentive observation regarding the tables

in the Supporting Information. The reviewer is correct, and we apologize for this oversight in the original submission.

4. *Figures and tables about applications in muscle and tendon/ligaments should also be included. In Figure 5, figures about blood vessels should also be included. Besides, if the authors want to include the tendon/ligaments in the manuscript, they should also mention it in the abstract and introduction sections.*

Authors' response: We sincerely thank the reviewer for these insightful suggestions regarding the figures, tables, and manuscript scope. We have implemented comprehensive revisions to address these points, which significantly enhance the manuscript.

Fig. S1. Application of mECM in tendon.

- A) Tendon hierarchical structure²².
- B) The mECM enhanced tenogenic differentiation²³.
- C) I) The mECM reduces senescence-associated β -galactosidase (SA- β -gal) activity. II-III) The mECM enhanced tenogenic differentiation. IV-VI) The mECM promotes the maintenance of stemness in aged tendon stem cells²⁴.

Fig. S2. Application of mECM in blood vessels.

- A) Blood vessels hierarchical structure²⁵.
- B) The mECM significantly promoted vascularization²⁶.
- C) The 3D mECM spheroids remain bioactive in vivo and promote angiogenesis. I-II) The

3D mECM spheroids promote the formation of endothelial. III) The 3D mECM spheroids promotes the formation of the nascent tubular structures formed by HUVECs within the constructs. IV) The 3D mECM spheroids promote angiogenesis²⁷.

5. *The English writing in the whole manuscript should be carefully checked, especially the use of singular and plural forms. Grammatical errors should be carefully corrected.*

Authors' response: We sincerely thank the reviewer for this crucial feedback. We acknowledge the importance of precise and polished language in a scientific review and apologize for the grammatical oversights in our initial submission.

References

1. Zhang, Y., Pizzute, T., Li, J., He, F. & Pei, M. sb203580 preconditioning recharges matrix-expanded human adult stem cells for chondrogenesis in an inflammatory environment - A feasible approach for autologous stem cell based osteoarthritic cartilage repair. *Biomaterials* **64**, 88-97 (2015).
2. Viji Babu, P. K., Rianna, C., Mirastschijski, U. & Radmacher, M. Nano-mechanical mapping of interdependent cell and ECM mechanics by AFM force spectroscopy. *Sci Rep* **9**, 12317 (2019).
3. Ragelle, H. *et al.* Comprehensive proteomic characterization of stem cell-derived extracellular matrices. *Biomaterials* **128**, 147-159 (2017).
4. Luan, D. *et al.* Cyclic Regulation of the Sulfilimine Bond in Peptides and NC1 Hexamers via the HOBr/H(2)Se Conjugated System. *Anal Chem* **90**, 9523-9528 (2018).
5. Benito-Jardón, M. *et al.* α v-Class integrin binding to fibronectin is solely mediated by RGD and unaffected by an RGE mutation. *J Cell Biol* **219** (2020).
6. Kollert, M. R. *et al.* Water and ions binding to extracellular matrix drives stress relaxation, aiding MRI detection of swelling-associated pathology. *Nat Biomed Eng* **9**, 772-786 (2025).
7. Li, X. *et al.* Studying the Effect of Receptors Clustering on Hyaluronic Acid Binding with CD44 and the Cell Entry of Hyaluronic Acid Targeting Nanodrugs at Single Molecule/Particle Level. *Anal Chem* **97**, 13048-13055 (2025).
8. Li, J. *et al.* Rejuvenation of chondrogenic potential in a young stem cell microenvironment. *Biomaterials* **35**, 642-653 (2014).
9. Cui, X. *et al.* Tissue-specific decellularized extracellular matrix rich in collagen, glycoproteins, and proteoglycans and its applications in advanced organoid engineering: A review. *Int J Biol Macromol* **315**, 144469 (2025).
10. Bae, J. Y., Park, S. Y., Shin, Y. H., Choi, S. W. & Kim, J. K. Preparation of human

- decellularized peripheral nerve allograft using amphoteric detergent and nuclease. *Neural Regen Res* **16**, 1890-1896 (2021).
11. Wang, Z. *et al.* Extracellular matrix derived from allogenic decellularized bone marrow mesenchymal stem cell sheets for the reconstruction of osteochondral defects in rabbits. *Acta Biomater* **118**, 54-68 (2020).
 12. Dikina, A. D., Almeida, H. V., Cao, M., Kelly, D. J. & Alsberg, E. Scaffolds Derived from ECM Produced by Chondrogenically Induced Human MSC Condensates Support Human MSC Chondrogenesis. *ACS Biomater Sci Eng* **3**, 1426-1436 (2017).
 13. Lu, H., Hoshiba, T., Kawazoe, N. & Chen, G. Comparison of decellularization techniques for preparation of extracellular matrix scaffolds derived from three-dimensional cell culture. *J Biomed Mater Res A* **100**, 2507-2516 (2012).
 14. Xiao, B. *et al.* Extracellular matrix from human umbilical cord-derived mesenchymal stem cells as a scaffold for peripheral nerve regeneration. *Neural Regen Res* **11**, 1172-1179 (2016).
 15. Shimizu, T. *et al.* The regeneration and augmentation of bone with injectable osteogenic cell sheet in a rat critical fracture healing model. *Injury* **46**, 1457-1464 (2015).
 16. Cai, R., Nakamoto, T., Kawazoe, N. & Chen, G. Influence of stepwise chondrogenesis-mimicking 3D extracellular matrix on chondrogenic differentiation of mesenchymal stem cells. *Biomaterials* **52**, 199-207 (2015).
 17. Jones, C. L., Penney, B. T. & Theodossiou, S. K. Engineering Cell-ECM-Material Interactions for Musculoskeletal Regeneration. *Bioengineering (Basel)* **10** (2023).
 18. Wang, Y. *et al.* Use of bioactive extracellular matrix fragments as a urethral bulking agent to treat stress urinary incontinence. *Acta Biomater* **117**, 156-166 (2020).
 19. Selman, M. & Pardo, A. Fibroageing: An ageing pathological feature driven by dysregulated extracellular matrix-cell mechanobiology. *Ageing Res Rev* **70**, 101393 (2021).
 20. Crapo, P. M., Gilbert, T. W. & Badylak, S. F. An overview of tissue and whole organ decellularization processes. *Biomaterials* **32**, 3233-3243 (2011).
 21. Gilbert, T. W., Sellaro, T. L. & Badylak, S. F. Decellularization of tissues and organs. *Biomaterials* **27**, 3675-3683 (2006).
 22. Zhu, G. *et al.* Bone physiological microenvironment and healing mechanism: Basis for future bone-tissue engineering scaffolds. *Bioact Mater* **6**, 4110-4140 (2021).
 23. Ning, L. J. *et al.* Constructing a highly bioactive tendon-regenerative scaffold by surface modification of tissue-specific stem cell-derived extracellular matrix. *Regen Biomater* **9**, rbac020 (2022).
 24. Jiang, D., Xu, B. & Gao, P. Effects of young extracellular matrix on the biological characteristics of aged tendon stem cells. *Adv Clin Exp Med* **27**, 1625-1630 (2018).
 25. Jiang, S., Wise, S. G., Kovacic, J. C., Rnjak-Kovacina, J. & Lord, M. S. Biomaterials containing extracellular matrix molecules as biomimetic next-generation vascular grafts. *Trends Biotechnol* **42**, 369-381 (2024).
 26. Ma, B., Wang, T., Li, J. & Wang, Q. Extracellular matrix derived from Wharton's Jelly-derived mesenchymal stem cells promotes angiogenesis via integrin $\alpha V\beta 3/c-$

- Myc/P300/VEGF. *Stem Cell Res Ther* **13**, 327 (2022).
27. Chiang, C. E. *et al.* Bioactive Decellularized Extracellular Matrix Derived from 3D Stem Cell Spheroids under Macromolecular Crowding Serves as a Scaffold for Tissue Engineering. *Adv Healthc Mater* **10**, e2100024 (2021).

REVIEWERS' COMMENTS:

Reviewer #1 (Remarks to the Author):

I thank the authors for addressing my questions and incorporating the necessary corrections into the text. At this stage, I have only a couple of comments:

1. *In sub-chapter 2.2 "Decellularizing the mECM", it appears that the authors omitted the description of chelating agents (provided in response to Reviewer #2's comment), which was mentioned in the Rebuttal Letter. Please make sure this is included in the main text. You are absolutely correct. We have already added the description of the use of chelating agents in the "2.2. Decellularizing the mECM" section.*

Authors' response: You are absolutely correct. We have now incorporated the description of chelating agent use into the "2.2. Decellularizing the mECM" section. Chelating agents, represented by ethylenediaminetetraacetic acid (EDTA) and ethylene glycol tetraacetic acid (EGTA), function by sequestering divalent cations (e.g., Ca^{2+} , Mg^{2+}) critical for maintaining cell membrane integrity and intercellular junctions¹. This cation depletion disrupts the structural stability of cell membranes, leading to gradual cell lysis, while their mild chemical nature minimizes damage to sensitive ECM components (e.g., collagen triple helices, glycosaminoglycan chains) that are vital for mECM's biomimetic activity². For instance, EDTA is frequently used to loosen cell adhesion to the ECM scaffold by chelating Ca^{2+} in focal adhesion complexes, and it is often combined with nucleases (DNase/RNase) to further eliminate residual genomic DNA and RNA—similar to the synergistic use of non-ionic detergents with nucleases. Notably, EGTA exhibits higher selectivity for Ca^{2+} over Mg^{2+} , making it preferable when preserving Mg^{2+} -dependent ECM enzymes (e.g., lysyl oxidase) is required for subsequent tissue regeneration¹.

2. *There are two tables labeled as Table S2 in the SUPPLEMENTAL MATERIAL. Kindly revise the table numbering and verify that all references to these tables in the main text are accurate.*

Authors' response: We sincerely apologize for this oversight. We have meticulously reviewed all tables in the Supplemental Material and corrected the numbering: The second instance labeled "Table S2" has been correctly renumbered to "Table S3". We have thoroughly checked the entire manuscript to ensure all in-text citations for "Table S2" and "Table S3" have been updated and now point to the correct tables. Furthermore, we have performed an additional check of all figure, table, section, and reference citations throughout the manuscript to prevent any similar numbering errors.

1. Gilbert, T. W., Sellaro, T. L. & Badylak, S. F. Decellularization of tissues and organs. *Biomaterials* **27**, 3675-3683 (2006).
2. Koo, M. A. *et al.* Preconditioning process for dermal tissue decellularization using electroporation with sonication. *Regen Biomater* **9**, rbab071 (2022).